# A molecular mechanism for the generation of ligand-dependent differential outputs by the epidermal growth factor receptor

Yongjian Huang[1,2,3], Jana Ognjenovic[4], Deepti Karandur[1,2,3], Kate Miller[1], Alan Merk[4], Sriram Subramaniam[5]*, John Kuriyan[1,2,3,6,7]*

[1]Department of Molecular and Cell Biology, University of California, Berkeley, Berkeley, United States; [2]Institute for Quantitative Biosciences, University of California, Berkeley, Berkeley, United States; [3]Howard Hughes Medical Institute, University of California, Berkeley, Berkeley, United States; [4]Frederick National Laboratory for Cancer Research, Frederick, United States; [5]University of British Columbia, Vancouver, Canada; [6]Department of Chemistry, University of California, Berkeley, Berkeley, United States; [7]Divisions of Molecular Biophysics and Integrated Bioimaging, Lawrence Berkeley National Laboratory, Berkeley, United States

*For correspondence:
Sriram.Subramaniam@ubc.ca
(SS);
kuriyan@berkeley.edu (JK)

**Abstract** The epidermal growth factor receptor (EGFR) is a receptor tyrosine kinase that couples the binding of extracellular ligands, such as EGF and transforming growth factor-α (TGF-α), to the initiation of intracellular signaling pathways. EGFR binds to EGF and TGF-α with similar affinity, but generates different signals from these ligands. To address the mechanistic basis of this phenomenon, we have carried out cryo-EM analyses of human EGFR bound to EGF and TGF-α. We show that the extracellular module adopts an ensemble of dimeric conformations when bound to either EGF or TGF-α. The two extreme states of this ensemble represent distinct ligand-bound quaternary structures in which the membrane-proximal tips of the extracellular module are either juxtaposed or separated. EGF and TGF-α differ in their ability to maintain the conformation with the membrane-proximal tips of the extracellular module separated, and this conformation is stabilized preferentially by an oncogenic EGFR mutation. Close proximity of the transmembrane helices at the junction with the extracellular module has been associated previously with increased EGFR activity. Our results show how EGFR can couple the binding of different ligands to differential modulation of this proximity, thereby suggesting a molecular mechanism for the generation of ligand-sensitive differential outputs in this receptor family.

## Editor's evaluation

This is an impressive study providing solid evidence for a molecular mechanism by which two related, high-affinity growth factors, binding in exactly the same site, can achieve differential signaling outputs through a dimerized receptor tyrosine kinase, and represents an important advance in the field.

## Introduction

Human EGFR can be activated by seven related ligands that generate different signaling outputs from the receptor. While the classical mechanism for receptor tyrosine kinase activation, ligand-induced

**Figure 1.** A structural collage of a ligand-bound EGFR dimer. Left, a schematic diagram of ligand-bound dimeric EGFR. Right, structures of EGFR fragments, determined previously: a dimeric extracellular module bound to EGF (PDB code: 3NJP *Lu et al., 2010*), a transmembrane helix dimer (PDB code: 2M20 *Endres et al., 2013*), and an asymmetric dimer of kinase domains (PDB code: 2GS6 *Zhang et al., 2006*).

dimerization, is essential for all these ligands to activate the receptor (*Kovacs et al., 2015*; *Leahy, 2004*; *Lemmon et al., 2014*), it cannot explain how the extracellular module of EGFR is able to respond differently to different ligands once the receptor is dimerized (*Ronan et al., 2016*; *Singh et al., 2016*; *Wilson et al., 2009*). The extracellular module of EGFR consists of four domains, denoted Domains I, II, III, and IV (*Figure 1*). Domains I and III sandwich the ligands between them (*Garrett et al., 2002*; *Ogiso et al., 2002*). Domain II, which bridges the ligand-binding domains, contains the 'dimerization arm', a loop that mediates the principal interaction with the other subunit in an activated dimer (*Dawson et al., 2005*). Domains I, II, and III form a compact unit that we refer to as the ligand-binding 'head', and Domain IV forms a rigid and elongated 'leg' that connects the ligand-binding head to the single transmembrane helix (*Figure 1*).

Previous crystallographic analyses of fragments of the EGFR extracellular module have revealed different conformations of the ligand-binding head when bound to EGF, TGF-α, epiregulin, and epigen (*Freed et al., 2017*; *Garrett et al., 2002*; *Lu et al., 2010*; *Ogiso et al., 2002*). It appeared that these conformational differences might explain how different EGFR ligands generate different signals (*Wilson et al., 2009*; *Scheck et al., 2012*), but the significance of these structural differences are unclear because only fragments of the receptor were analyzed. For weakly binding ligands, such as epiregulin and epigen, the lifetime of the dimeric receptor is shortened relative to that for the tighter-binding ligands, such as EGF, and the reduction in dimer lifetime affects signaling output (*Freed et al., 2017*). However, EGF and TGF-α are both high-affinity ligands (*Jones et al., 1999*), and they also generate different outputs from EGFR. How this happens is still unclear.

Cell-based studies using fluorescent reporters of EGFR conformation have shown that the binding of EGF and TGF-α results in different conformations of the intracellular juxtamembrane segments of the receptor (*Scheck et al., 2012*; *Doerner et al., 2015*; *Lowder et al., 2015*). The balance between

alternative conformations of the juxtamembrane segments can be modulated by point mutations in the transmembrane helices (*Sinclair et al., 2018*) and oncogenic mutations in the kinase domains (*Lowder et al., 2015*). A more recent study reveals that simply switching the conformation of the juxtamembrane segments by mutation suffices to switch EGFR signaling outputs (*Mozumdar et al., 2021*). These studies demonstrate that the extracellular module, the transmembrane helix, the intracellular juxtamembrane segments, and the intracellular kinase domains are coupled allosterically, and that they respond differently to EGF and TGF-α. How the two ligands generate these differences remains unknown. To gain structural insight into this mechanism, we analyzed full-length human EGFR in complex with EGF and TGF-α by cryo-EM.

## Results and discussion

### Linkages between the extracellular modules and the transmembrane helices are not rigidly defined

We purified full-length human EGFR expressed in a human HEK293S GnTI⁻ cell line (*Reeves et al., 2002*; *Qiu et al., 2009*), and reconstituted the complex of EGFR with EGF into four membrane-mimetic environments: detergent micelles, lipid nanodiscs (*Bayburt and Sligar, 2010*), amphipols (*Gohon et al., 2006*), and peptidiscs (*Angiulli et al., 2020*; *Carlson et al., 2018*; *Figure 2—figure supplement 1*). Cryo-EM reconstructions of the full-length EGFR:EGF complex in all four reconstitutions look very similar, and only the extracellular modules are resolved in the cryo-EM densities (*Figure 2*). The lack of resolvable density for the transmembrane helices and the intracellular kinase domains suggests that the linkage between the extracellular modules and the transmembrane helices is not rigidly defined, consistent with 2D class averages of negative-stain EM images (*Mi et al., 2011*; *Figure 2—figure supplement 1*). Thus, mechanisms for ligand-dependent differential responses in EGFR that require strict stereochemical coupling between the extracellular ligand-binding modules and the transmembrane helices appear to be ruled out. Recently published cryo-EM analyses of intact insulin receptor and intact type 1 insulin-like growth factor receptor also did not resolve the transmembrane and intracellular modules (*Li et al., 2019*; *Uchikawa et al., 2019*; *Zhang et al., 2020*), suggesting that a lack of rigidity in the connection between the extracellular module and the transmembrane helices is a feature common to receptor tyrosine kinases.

### Cryo-EM single-particle analysis reveals a range of conformations for the EGF-bound extracellular module

After focusing the cryo-EM reconstructions on the extracellular module alone, the dataset for the full-length EGFR:EGF complex reconstituted in peptidiscs yielded the highest resolution reconstruction of the dimeric extracellular module, with an overall resolution of 2.9 Å (*Figure 3A*, *Figure 3—source data 1*; *Figure 3—video 1*). In this reconstruction, cryo-EM density for the Domain IV legs is less well resolved than for the heads. To better understand the origin of this apparent structural heterogeneity, we used 3D classification to further partition the entire dataset of 765,883 particles into 10 subclasses (*Punjani et al., 2017*). Cryo-EM reconstructions corresponding to individual subclasses resulted in overall resolutions ranging between 3.1 Å and 3.7 Å (*Figure 3A*, *Figure 3—figure supplement 1*).

The resulting structures for different subclasses differ from one another in the extent of a 'scissor-like' rotation between the two ligand-binding heads of the dimeric extracellular module (*Figure 3A*, *Figure 3—videos 1–3*). The rotation between the two ligand-binding heads is due to a rigid-body rotation of each head about an axis parallel to the dimerization arms, and is correlated with a slight twist in the dimerization arm of each EGFR subunit (*Figure 3B–D*, *Figure 3—videos 1–3*). The extent of this intersubunit rotation can be estimated by defining a dihedral angle corresponding to rotation about a virtual bond connecting the Cα atoms of Thr 249 in each EGFR subunit (we number amino acid residues in EGFR without including the 24-residue signal sequence). The positions of the Cα atoms of Ile 190 in each subunit are used to further specify the dihedral angle, and the value of this dihedral angle ranges from ~10° to ~25° in the 10 representative structures (*Figure 3A*).

The scissor-like rotation between ligand-binding heads in the cryo-EM structures is reminiscent of differences noted previously between crystal structures of fragments of EGFR extracellular modules bound to different ligands (*Liu et al., 2012*). At one extreme, the EM-derived structure with the largest intersubunit rotation (~25°) resembles closely the crystal structure of the ligand-binding head

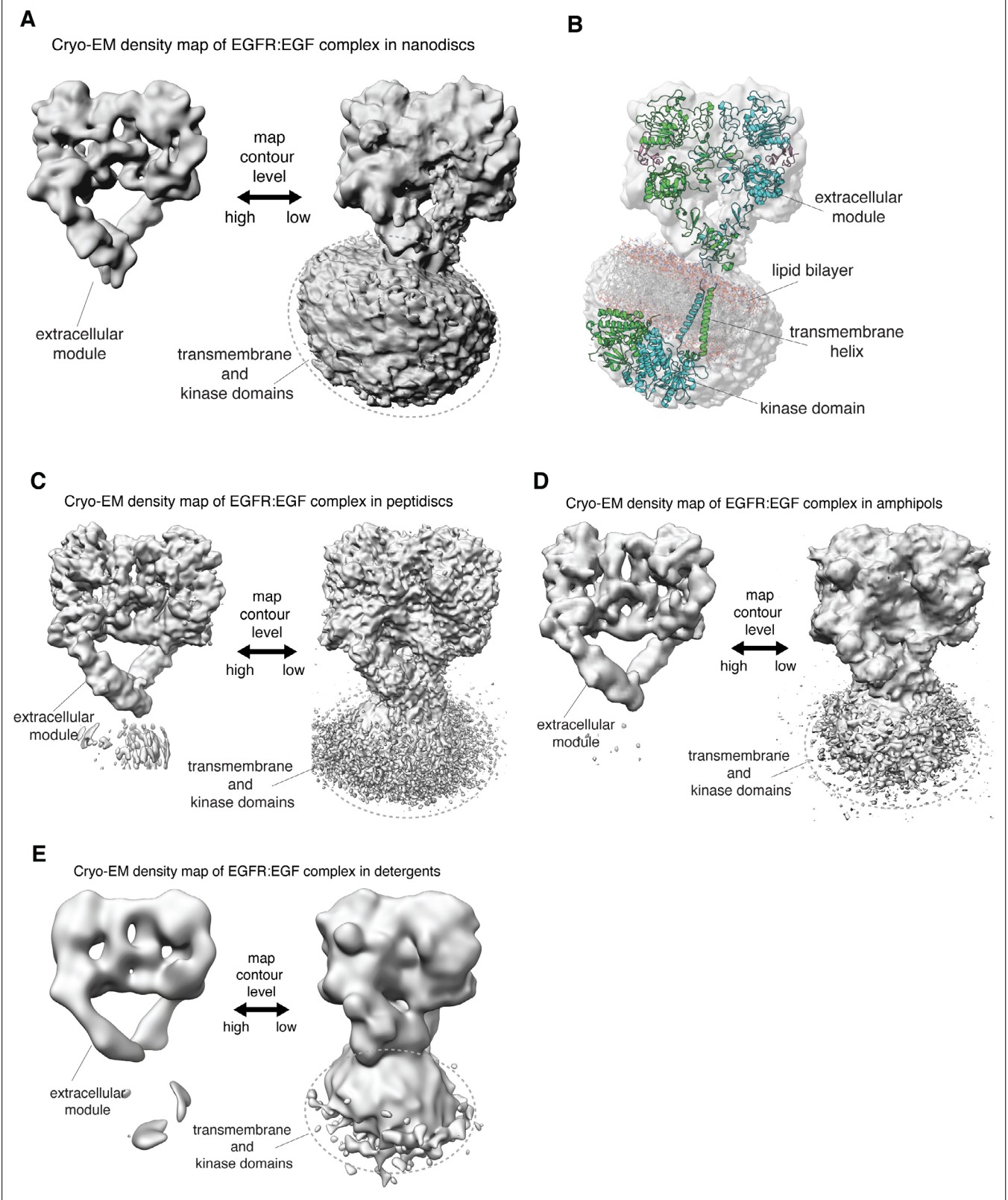

**Figure 2.** Linkage between the extracellular module and transmembrane helices in full-length human EGFR is not rigid. (**A, C–E**) In each panel, the cryo-EM density map for a reconstituted full-length EGFR:EGF complex is shown at two contour levels. Note the lack of EM density for the transmembrane and kinase domains of EGFR at higher contour level (left) in all these complexes. (**A**) Full-length EGFR:EGF complex reconstituted in nanodiscs. Map shown at two contour levels, high (σ: ~ 5.8) and low (σ:~0.6). (**B**) Superposition of the cryo-EM map shown in **A** with a model for full-length EGFR

*Figure 2 continued on next page*

*Figure 2 continued*

embedded in the lipid bilayer, described previously (***Arkhipov et al., 2013***). (**C**) Full-length EGFR:EGF complex reconstituted in peptidiscs. Map shown at two contour levels, high (σ: ~ 5.9) and low (σ:~0.8). (**D**) Full-length EGFR:EGF complex reconstituted in amphipols. Map shown at two contour levels, high (σ: ~ 12.0) and low (σ:~1.2). (**E**) Full-length EGFR:EGF complex reconstituted in detergents. Map shown at two contour levels, high (σ: ~ 12.8) and low (σ:~1.0).

The online version of this article includes the following figure supplement(s) for figure 2:

**Figure supplement 1.** Reconstitution of full-length human EGFR:EGF complex.

of EGFR bound to EGF (PDB code: 3NJP (***Lu et al., 2010***; ***Ogiso et al., 2002***), root-mean-square deviations (RMSDs) in Cα positions calculated over EGFR residues 1–501: 0.8 Å, ***Figure 4A***). The EM-derived structure with the smallest intersubunit rotational angle (~10°) resembles the crystal structure of the EGFR ligand-binding head bound to TGF-α (PDB code: 1MOX (***Garrett et al., 2002***), RMSDs in Cα positions calculated over EGFR residues 1–501: 1.5 Å, ***Figure 4B***). Thus, the partitioning of the cryo-EM data into subclasses reflects an intrinsic conformational variability of the ligand-bound EGFR extracellular module. Importantly, the cryoEM results demonstrate that the conformations ascribed previously to specific binding of EGF or TGF-α to the receptor are actually part of an ensemble of conformations that are readily accessible to EGFR when bound to either one of these ligands.

## Different dimeric conformations of the ligand-bound extracellular module have different separations between the tips of the two legs

The two Domain IV legs in the dimer behave differently when the scissor-like rotational angle between the ligand-binding heads changes. The structure of the Domain IV leg from a previously determined crystal structure of the EGFR extracellular module (PDB code: 3NJP ***Lu et al., 2010***) can be docked into our cryo-EM densities (***Figure 3—figure supplement 2A, B***). This allowed us to determine the position and orientation of the Domain IV legs associated with each representative conformation from the ensemble revealed by cryo-EM. When the rotational angle between the two heads is at its largest value (~25°), the two legs adopt similar conformations, and the membrane-proximal tips of the two legs are juxtaposed, with a separation of ~5 Å between the Cαatoms of the last residue in the legs (Thr 614) (the 'juxtaposed' conformation, ***Figures 3B and 5***). When the rotation between the two subunits is small, maintenance of the same conformation in both legs would lead to steric clashes between the two legs (***Figure 3—figure supplement 2C***). The clashes are relieved by one of the two Domain IV legs undergoing a hinge rotation around the connection between the head and the leg, and we refer to this leg as 'flexible'. This hinge rotation causes the flexible leg to swing upwards, closer to the ligand-binding head, increasing the separation between the membrane-proximal tips of the two Domain IV legs in the dimer. When the angle between the heads is at its minimal value (~10°), the flexible leg swings maximally toward the ligand-binding head, increasing the distance between the membrane-proximal tips of the extracellular module by nearly 15 Å (the 'separated' conformation, ***Figures 3D and 5***).

## The tips-separated conformation of the dimeric extracellular module can couple more readily to the N-terminal dimer of the transmembrane helices

The EGFR transmembrane helix contains two dimerization motifs, one located toward the N-terminal end of the transmembrane helix, near the extracellular face of membrane, and the other located toward the C-terminal end (***Mendrola et al., 2002***). These dimerization motifs are involved in close association between the transmembrane helices in alternative structures defined by NMR, in which the helices are splayed apart at either the N-terminal or the C-terminal ends (***Bocharov et al., 2016***; ***Bocharov et al., 2017***; ***Endres et al., 2013***). To understand how these alternative dimeric configurations of the transmembrane helices connect to the different conformations of the extracellular module, with different separation between the tips of the Domain IV legs, we analyzed two structures of EGFR transmembrane helix dimers determined previously by NMR. One structure shows N-terminal association of the transmembrane helices (PDB code: 5LV6 ***Bocharov et al., 2017***) and the other shows C-terminal association (PDB code: 2M0B ***Bocharov et al., 2016***; ***Figure 6A***).

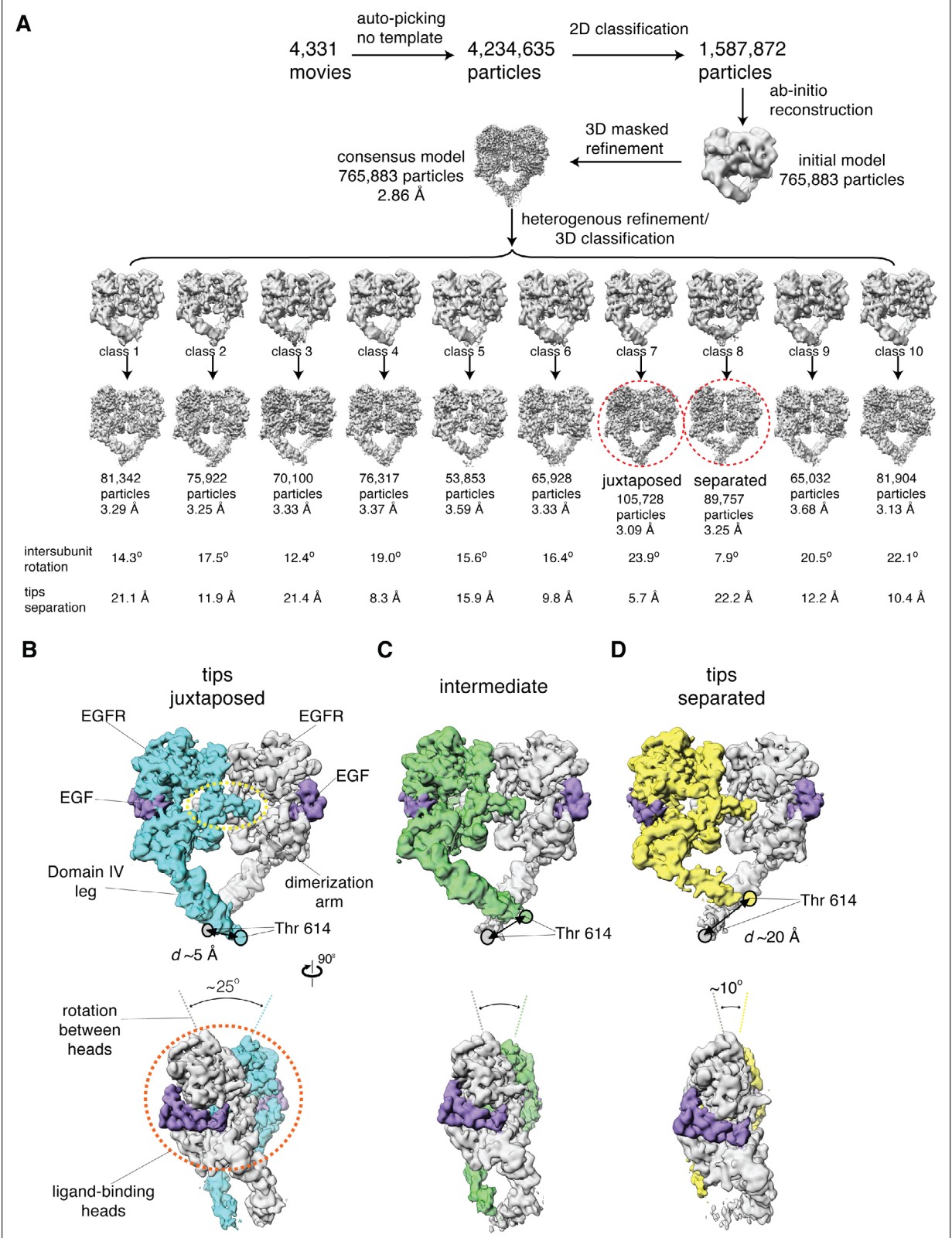

**Figure 3.** Cryo-EM analysis reveals a range of conformations for the EGF-bound extracellular module. (**A**) Summary of the cryo-EM data processing workflow for the full-length EGFR:EGF complex reconstituted in peptidiscs. The two extreme conformations of the ensemble, the juxtaposed and the separated, are highlighted by red dotted circle. The intersubunit rotational angles and the separations between the tips of the extracellular module are summarized below each subclass. See Materials and methods for more details. (**B–D**) Two orthogonal views of the cryo-EM maps for three

*Figure 3 continued on next page*

*Figure 3 continued*

representative states of the conformational ensemble of the EGF-bound extracellular module: the tips-juxtaposed conformation (**B**), an intermediate conformation (**C**), and the tips-separated conformation (**D**). These conformations differ from one another in the rotations between the ligand-binding heads, as well as in the distances between the two Thr 614 residues located at the membrane-proximal ends of the Domain IV legs.

The online version of this article includes the following video and figure supplement(s) for figure 3:

**Source data 1.** Statistics of cryo-EM data collection and structure determination.

**Figure supplement 1.** Cryo-EM image processing for focused reconstruction on the extracellular module of full-length EGFR:EGF complex reconstituted in peptidiscs.

**Figure supplement 2.** Conformation of the Domain IV leg.

**Figure supplement 3.** Cryo-EM data processing workflow.

**Figure 3—video 1.** 3D variability analysis of the EGFR:EGF complex performed in cryoSPARC v2 (*Punjani and Fleet, 2020*), shown from the front. https://elifesciences.org/articles/73218/figures#fig3video1

**Figure 3—video 2.** 3D variability analysis of the EGFR:EGF complex performed in cryoSPARC v2 (*Punjani and Fleet, 2020*), shown from the side. https://elifesciences.org/articles/73218/figures#fig3video2

**Figure 3—video 3.** 3D variability analysis of the EGFR:EGF complex performed in cryoSPARC v2 (*Punjani and Fleet, 2020*), shown from the top. https://elifesciences.org/articles/73218/figures#fig3video3

The seven-residue linker connecting the extracellular module and the transmembrane helices (Asn 615-Ser 621) is modeled in both NMR structures. The three C-terminal residues of the linker (Ile 619-Ser 621), located at the junction with the transmembrane helix, adopt well-defined conformations in both NMR structures. The hydrophobic sidechain of Ile 619 packs against the sidechain of Ala 623 in the first turn of the transmembrane helix. Ile 619 is followed by a $3_{10}$-helical turn spanning Pro 620 to Ala 623 (*Figure 6B*). These structural constraints cause the linker-transmembrane helix junction to orient parallel to the membrane surface in both structures (*Figure 6A*). As a consequence, in the N-terminal dimer of the transmembrane helices, the two linkers are oriented away from each other, with the two Ile 619 residues separated by ~20 Å. In contrast, the C-terminal dimer has the two linkers oriented toward each other, with the distance between the two Ile 619 residues at ~10 Å. Thr 614 is at the junction between Domain IV leg of the extracellular module and the linker. In the ensemble of NMR structures for the N-terminal dimer of the transmembrane helices (*Bocharov et al., 2017*), the two threonine residues (Thr 614) are separated by an average distance of ~35 Å. In comparison, the separation between the two threonine residues is ~20 Å in the ensemble of structures for the C-terminal dimer of the transmembrane helices (*Bocharov et al., 2016*; *Figure 6A*). These observations suggest that the junctions between the extracellular module and the transmembrane helices are more separated in the N-terminal dimer of the transmembrane helices than the connection in the C-terminal dimer.

We used molecular dynamics simulations to explore the constraints on the separation between the two threonine residues at position 614 in the linker, starting either the N-terminal dimer configuration (PDB code: 5LV6 *Bocharov et al., 2017*) or the C-terminal dimer configuration (PDB code: 2M0B *Bocharov et al., 2016*). Following equilibration (see Materials and methods for full details), we initiated 25 independent molecular dynamics trajectories for both configurations, each 100 ns long, for a total simulation time of 2.5 µs for each configuration. The configuration of the transmembrane helices remains close to the starting configuration over this timescale. The conformation of the junction between the linker and the transmembrane helices is also very stable in both sets of simulations, with the sidechain of Ile 619 remaining associated with Ala 623 over the course of the simulations (*Figure 6E and F*). In the simulations of the N-terminal dimer configuration, the linkers explore a range of configurations but are oriented predominantly such that they point away from one another, and the distance between Thr 614 from each subunit ranges between 20 Å and 35 Å (*Figure 6C*). In the simulations starting from the C-terminal dimer configuration, the Thr 614 residues from each subunit are much closer to each other, predominantly sampling distances between 7 Å and 12 Å (*Figure 6C*). These results further support the idea that the junction between the linker and the transmembrane helices imposes a constraint on the orientation of the linkers. When the transmembrane helices are in the N-terminal dimer configuration, this linker favors a connection to the tips-separated conformation of the extracellular module, and when the transmembrane helices are in the C-terminal dimer configuration, a connection to the tips-juxtaposed conformation of the extracellular module is indicated.

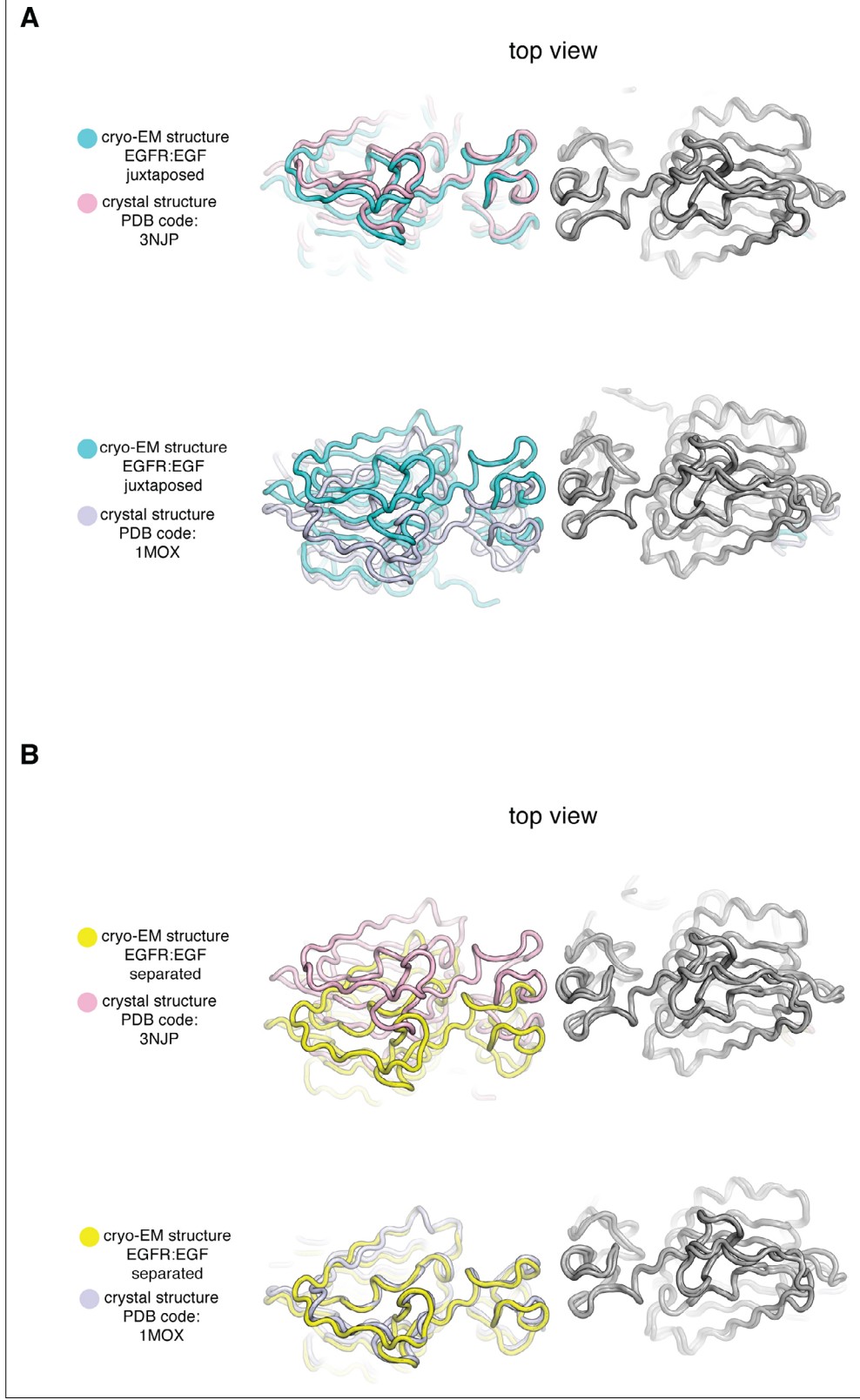

**Figure 4.** Comparison of Cryo-EM structures of EGFR:EGF complex in two extreme conformations with previous crystal structures of the ligand-binding head of EGFR bound to EGF and TGF-α. In each panel, a cryo-EM structure of the extracellular module of the full-length EGFR:EGF complex (juxtaposed or separated) is compared to a crystal structure of the ligand-binding head of EGFR bound to EGF or TGF-α. In each comparison, structures are

*Figure 4 continued on next page*

*Figure 4 continued*

superimposed on the right-hand EGFR subunits (gray). (**A**) Top, the cryo-EM structure of juxtaposed conformation of EGFR:EGF (cyan) is compared to a crystal structure of the ligand-binding head of EGFR bound to EGF (pink, PDB code: 3NJP). Bottom, the cryo-EM structure of juxtaposed conformation of EGFR:EGF (cyan) is compared to a crystal structure of the ligand-binding head of EGFR bound to TGF-α (light purple, PDB code: 1MOX). Note that the cryo-EM structure of the juxtaposed conformation of the EGFR:EGF complex resembles closely the crystal structure of the ligand-binding head of EGFR bound to EGF (PDB code: 3NJP), but deviates from the crystal structure of the ligand-binding head of EGFR bound to TGF-α (PDB code: 1MOX). (**B**) Top, the cryo-EM structure of separated conformation of EGFR:EGF (yellow) is compared to a crystal structure of the ligand-binding head of EGFR bound to EGF (pink, PDB code: 3NJP). Bottom, the cryo-EM structure of separated conformation of EGFR:EGF (yellow) is compared to a crystal structure of the ligand-binding head of EGFR bound to TGF-α (light purple, PDB code: 1MOX). Note that the cryo-EM structure of the separated conformation of the EGFR:EGF complex resembles closely the crystal structure of the ligand-binding head of EGFR bound to TGF-α (PDB code: 1MOX), but deviates from the crystal structure of the ligand-binding head of EGFR bound to EGF (PDB code: 3NJP).

We computed the free-energy change of the system when the distance between the Thr 614 residues from each subunit is driven from 27 Å (the peak in the distribution of distances between these residues in simulations of the N-terminal dimer, see *Figure 6C*) to 8 Å (the corresponding peak in simulations of the C-terminal dimer, see *Figure 6C*), when the transmembrane helices are in the N-terminal dimer configuration. We used umbrella-sampling simulations (*Torrie and Valleau, 1977*), with sampling windows spaced 1 Å apart (for a total of 20 windows) and starting structures for each window taken from a steered molecular dynamics simulation. Three independent umbrella-sampling calculations were performed, with the starting structures for each taken from different steered molecular dynamics trajectories. These calculations converged (i.e. the potential of mean force does not change) when each window sampled ~60–90 ns of simulation. All three umbrella-sampling simulations yield similar results (*Figure 6D*), with the free-energy being lowest when the distance between the Cα atoms of Thr 614 residues from each subunit is ~26 Å. Decreasing this distance to 8 Å leads to the free energy of the system increasing by ~9 kJ.mol$^{-1}$ (~3.5 $k_B T$ at room temperature, *Figure 6D*). This indicates that in the N-terminal dimer configuration, it is thermodynamically unfavorable for the linkers between the extracellular modules and the transmembrane helices to sample conformations where the distance between the Thr 614 residues is small. Thus, when the transmembrane helices are in the N-terminal dimer configuration, it is likely that the tips-separated conformation of the extracellular module dimer, with the tips of Domain IV at a larger distance, is favored over the tips-juxtaposed conformation.

## An oncogenic mutation in the kinase domain stabilizes the tips-separated conformation of the extracellular module

Dimerization of the transmembrane helices through the N-terminal dimerization motif is associated with increased kinase activity (*Miloso et al., 1995*; *Lu et al., 2010*; *Pahuja et al., 2018*; *Sinclair et al., 2018*), so our structural analysis of the connection between the extracellular module and the transmembrane helices suggests that the tips-separated conformation of the extracellular module is associated with a more active conformation of the kinase domains, through its coupling to the N-terminal dimer of the transmembrane helices.

To test this hypothesis, we examined an oncogenic variant of EGFR seen frequently in cancer patients, in which a leucine residue in the activation loop of the intracellular kinase domain is substituted by arginine (L834R; this is residue 858 when the signal sequence is included). The activation loop is a key allosteric control element at the active site of the kinase domain, and the L834R mutation promotes the formation of the catalytically active asymmetric dimer of the kinase domains, leading to hyper-activation of EGFR (*Red Brewer et al., 2013*; *Shan et al., 2012*). We verified that this mutation increases the activity of EGFR (*Figure 9—figure supplement 1D*), consistent with previous results (*Red Brewer et al., 2013*). We carried out cryo-EM analysis of full-length EGFR(L834R) bound to EGF, using the same procedures as for wild-type EGFR (*Figure 3—figure supplement 3A*). The two structures representing the tips-juxtaposed and the tips-separated conformations of the EGFR(L834R):EGF complex were determined at resolution of 3.3 Å and 3.4 Å, respectively (*Figure 3—source data 1*).

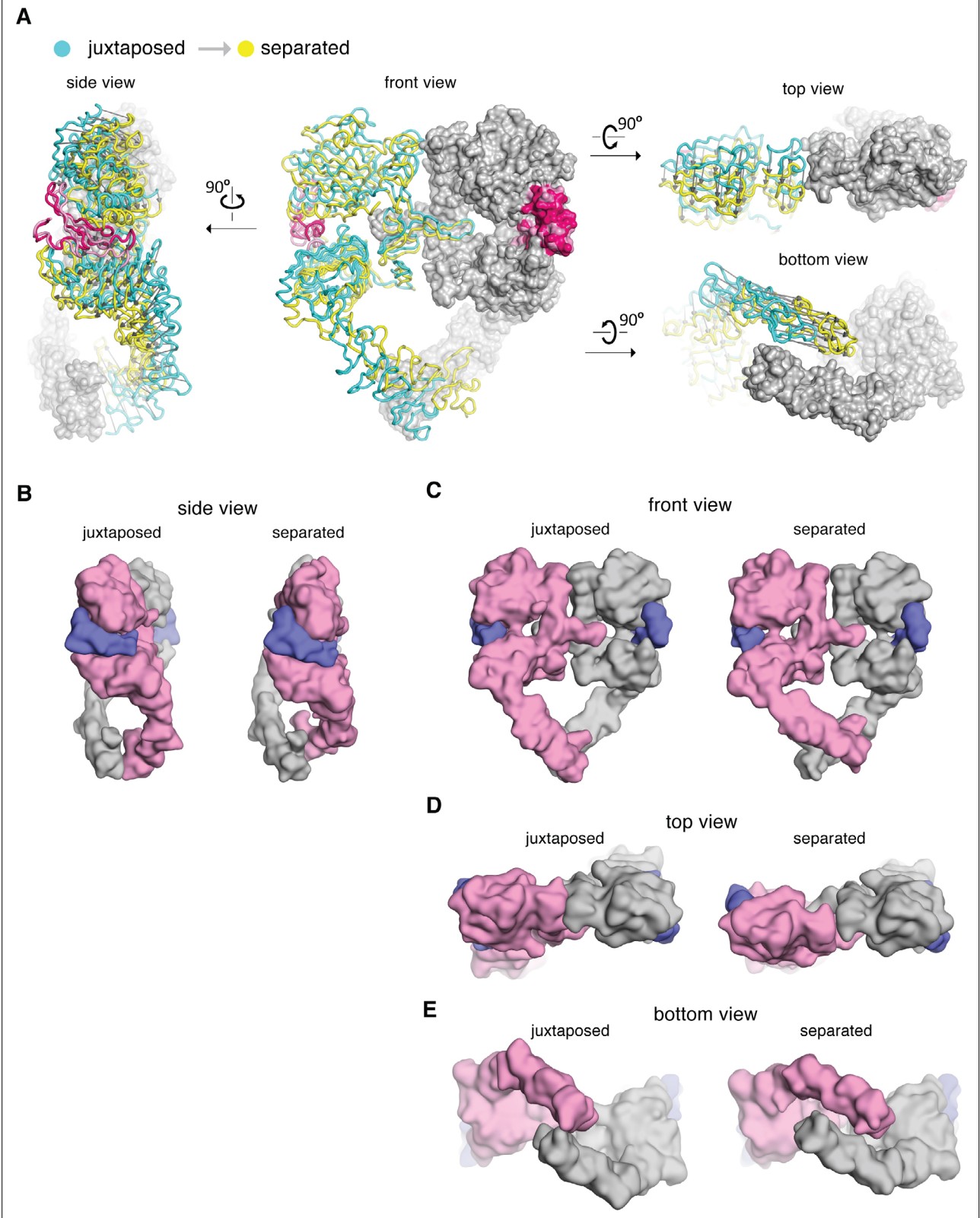

**Figure 5.** Comparison between the cryo-EM structures of the juxtaposed and the separated conformations of the extracellular module of the EGFR:EGF complex. (**A**) The cryo-EM structures representing the juxtaposed conformation (cyan) and the separated conformation (yellow) are superimposed on the right-hand EGFR subunits which have similar Domain IV leg conformations (light gray). A vector (dark gray) is used to indicate the displacement of Cα atoms between the two structures, pointing from the juxtaposed structure to the separated structure. (**B–E**) Side-by-side comparison between the

*Figure 5 continued on next page*

Figure 5 continued

juxtaposed conformation and the separated conformation of the extracellular module of the EGFR:EGF complex. Molecular surface of the cryo-EM models of the juxtaposed and separated conformations are shown from the side (B), the front (C), the top (D), and the bottom (E). EGF molecules are shown in slate blue. The same EGFR subunit used for superposition in (**A**) is colored gray and the other EGFR subunit is light pink.

The overall structures of the tips-juxtaposed and tips-separated conformations of the EGFR(L-834R):EGF complex are very similar to those seen for the corresponding complex of the wild-type receptor. The effect of the activating mutation, L834R, becomes apparent when comparing the quality of the cryo-EM densities in the Domain IV legs for the wild-type and mutant EGFR. For the tips-juxtaposed conformations, the cryo-EM densities are of comparable quality (*Figure 7*). An important result emerges when comparing the tips-separated conformations for wild-type EGFR and the oncogenic variant. Although the oncogenic L834R mutation involves a residue that is located in the intracellular kinase domain, this mutation markedly stabilizes the flexible leg in the tips-separated conformation of the extracellular module on the other side of the membrane, compared to wild-type EGFR. The cryo-EM density for the flexible leg is much better resolved in the EGFR(L834R):EGF complex than for wild-type EGFR, with secondary structural features clearly visible (*Figure 7*). Thus, activation of the intracellular kinase domains by oncogenic mutation preferentially stabilizes the flexible leg in the tips-separated conformation of the extracellular module, providing direct evidence for conformational coupling between the extracellular and intracellular modules of EGFR. These results are in agreement with the idea that the tips-separated conformation of the extracellular module is coupled to the more active conformations of the transmembrane helices and the intracellular kinase domains.

## EGF and TGF-α differ in their ability to stabilize the flexible domain IV leg in the tips-separated conformation

Given the association of the tips-separated conformation of the extracellular module with increased activity of the receptor, we wondered if EGF and TGF-α can differentially modulate different conformations of the extracellular module. We carried out cryo-EM analysis of EGFR bound to TGF-α using the same procedures as before (*Figure 3—figure supplement 3B*). Compared to the EGFR:EGF complex, the flexible leg in the tips-separated conformation of the EGFR:TGF-α complex is destabilized. The cryo-EM density for the flexible leg is poorly defined, and there is no interpretable density for this leg in the reconstruction of the tips-separated conformation of the EGFR:TGF-α complex (*Figure 7*). For the tips-juxtaposed conformation, the reconstructions for the EGFR:EGF and EGFR:T-GF-α complexes are similar (*Figure 7*), with near-symmetrical densities for the Domain IV legs in both complexes. The only noticeable difference is that the cryo-EM density at the very end of the Domain IV legs in the EGFR:TGF-α complex is slightly weaker than the corresponding density in the EGFR:EGF complex (*Figure 7*).

The structures we have determined provide a plausible explanation for why EGF and TGF-α differ in their ability to stabilize the flexible leg in the tips-separated conformation. Several key interactions with Domain I of the receptor are different between EGF and TGF-α (*Figure 8A*). As a result, the ligand-binding pocket is slightly compressed when TGF-α is bound. We used molecular dynamics simulations to investigate this difference. We initiated four trajectories from the crystal structure of the EGFR extracellular module bound to EGF (PDB code: 3NJP *Lu et al., 2010*). We also initiated four trajectories starting from this crystal structure (PDB code: 3NJP), but with the EGF replaced by TGF-α, with the structure of the ligand taken from the crystal structure of truncated EGFR bound to TGF-α (PDB code: 1MOX *Garrett et al., 2002*). Each trajectory was run for between ~350 and ~ 550 ns, for a total simulation time of ~1.8 μs for EGFR bound to EGF and ~2.0 μs for EGFR bound to TGF-α. We measured the angle between the centroids of Domain I, part of Domain II and Domain III (*Figure 8B*) over the course of the simulations. In EGFR bound to EGF, this angle is slightly larger, with the mode occurring at ~80°. In the simulations of EGFR bound to TGF-α, this angle is slightly smaller, with the mode occurring at ~75°, indicating that the ligand-binding pocket is slightly compressed when EGFR is bound to TGF-α (*Figure 8C*). This compression between Domains I and III causes Domain II to bend in the TGF-α complex relative to its structure in the EGF complex (*Figure 8A*), leading to a small difference in the orientation of the dimerization arm with respect to Domains III and IV. EGF and TGF-α both maintain the same interface between the dimerization arms of the two EGFR subunits in the dimer, but

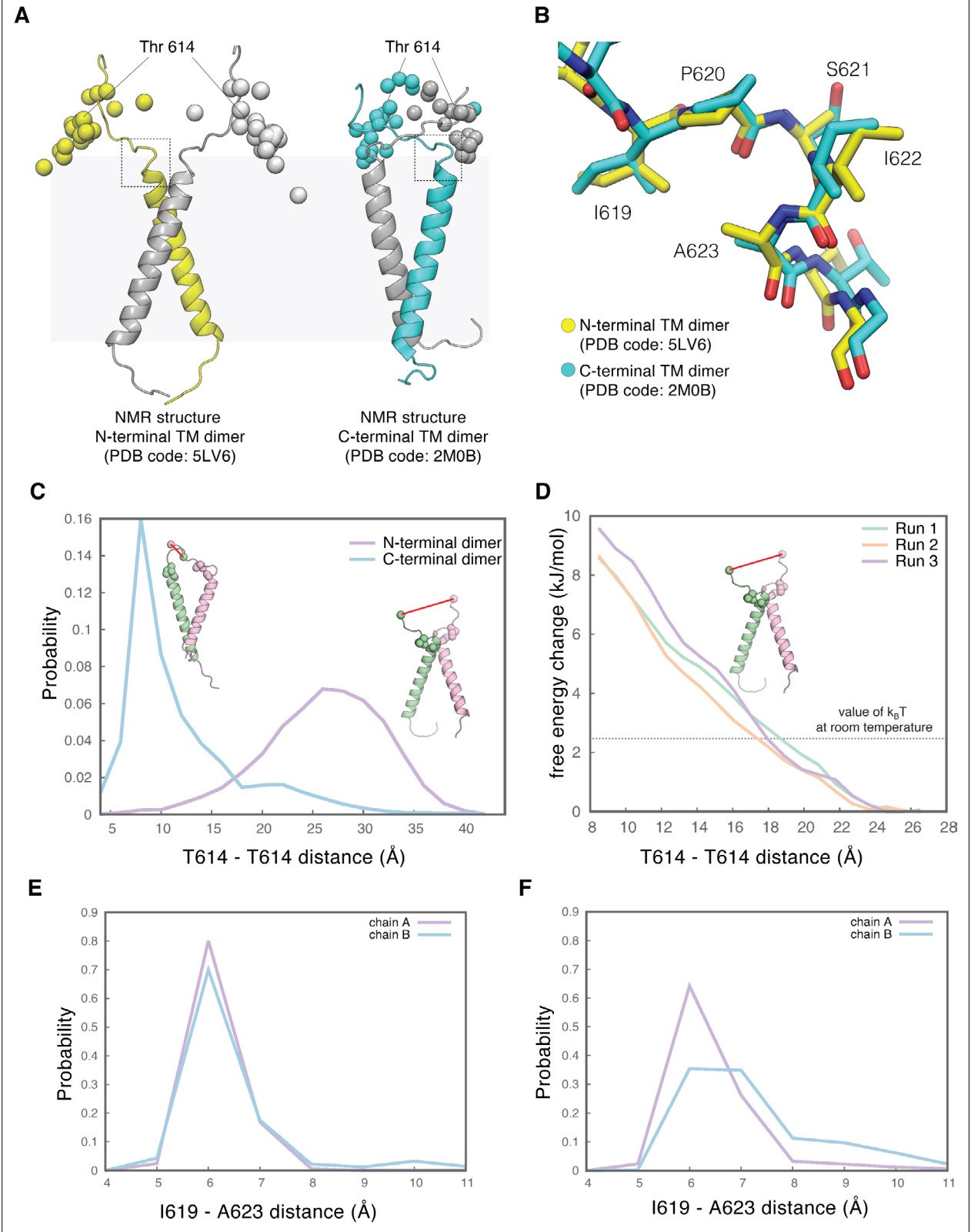

**Figure 6.** The tips-separated conformation of the extracellular module can couple more readily to the N-terminal dimer of the transmembrane helices. (**A**) Models of the transmembrane helices in two configurations based on previous NMR structures. One of the structures in the NMR-based ensemble for the N-terminal dimer configuration (PDB code: 5LV6) is shown in yellow (chain A) and gray (chain B), and the structure for the C-terminal dimer configuration (PDB code: 2M0B) is shown in cyan (chain A) and gray (chain B). Shown in spheres are superimposed Cα atoms of the residue Thr 614,

*Figure 6 continued on next page*

*Figure 6 continued*

located at the end of the extracellular module, from the 20 structures in the NMR-based ensemble for the N-terminal dimer configuration (left) and for the C-terminal dimer configuration (right), respectively. The black dotted square indicates the linker region that is imposing a constraint on the orientation of the linkers. In the N-terminal dimer configuration of the transmembrane helices (left), the linkers from each subunit are oriented away from one another. In the C-terminal dimer configuration of the transmembrane helices (right), the linkers are oriented toward each other. (**B**) Enlarged view of the linker located at the junction with the transmembrane helices, zoomed in from the areas highlighted by dotted black square in **A**, showing well-defined conformations for this region in both NMR structures. (**C**), Probability distribution of the distances between the Cα atoms of the two Thr 614 residues, sampled every 250 picoseconds over 2.5 μs of MD simulations of the N-terminal dimer configuration (purple) and the C-terminal dimer configuration (cyan) of the transmembrane helices. Initial structures of the transmembrane helices dimers with the linker are shown as cartoons. Residues Thr 614 located at the end of the extracellular module are shown as spheres. Also shown as spheres are residues Ile 619 and Ala 623, located at the junction between the linker and the transmembrane helix, and the interaction between these two residues is maintained over 2.5 μs MD simulations (see **E** and **F**). (**D**) Free-energy change of the system when the distance between the Thr 614 residues from each subunit is driven from 27 Å to 8 Å, when the transmembrane helices are in the N-terminal dimer configuration. Results from three independent umbrella-sampling calculations are shown in green, orange, and purple. The dotted gray line indicates the value of $k_B$T at room temperature, 2.479 kJ/mol. (**E and F**) Probability distribution of the distances between the Cα atoms of Ile 619 and Ala 623, sampled every 250 picoseconds over 2.5 μs of MD simulations of the N-terminal dimer configuration (**E**) and the C-terminal dimer configuration of the transmembrane helices (**F**).

the additional bending of Domain II in the TGF-α complex requires the two Domain IV legs to rotate toward each other. In the tips-separated conformation of the TGF-α complex, this inward motion would cause a steric clash between the two legs if the legs maintained the conformation seen in the EGF complex (*Figure 9A*). The increased disorder in the Domain IV leg when TGF-α is bound appears to be a result of this close contact being avoided.

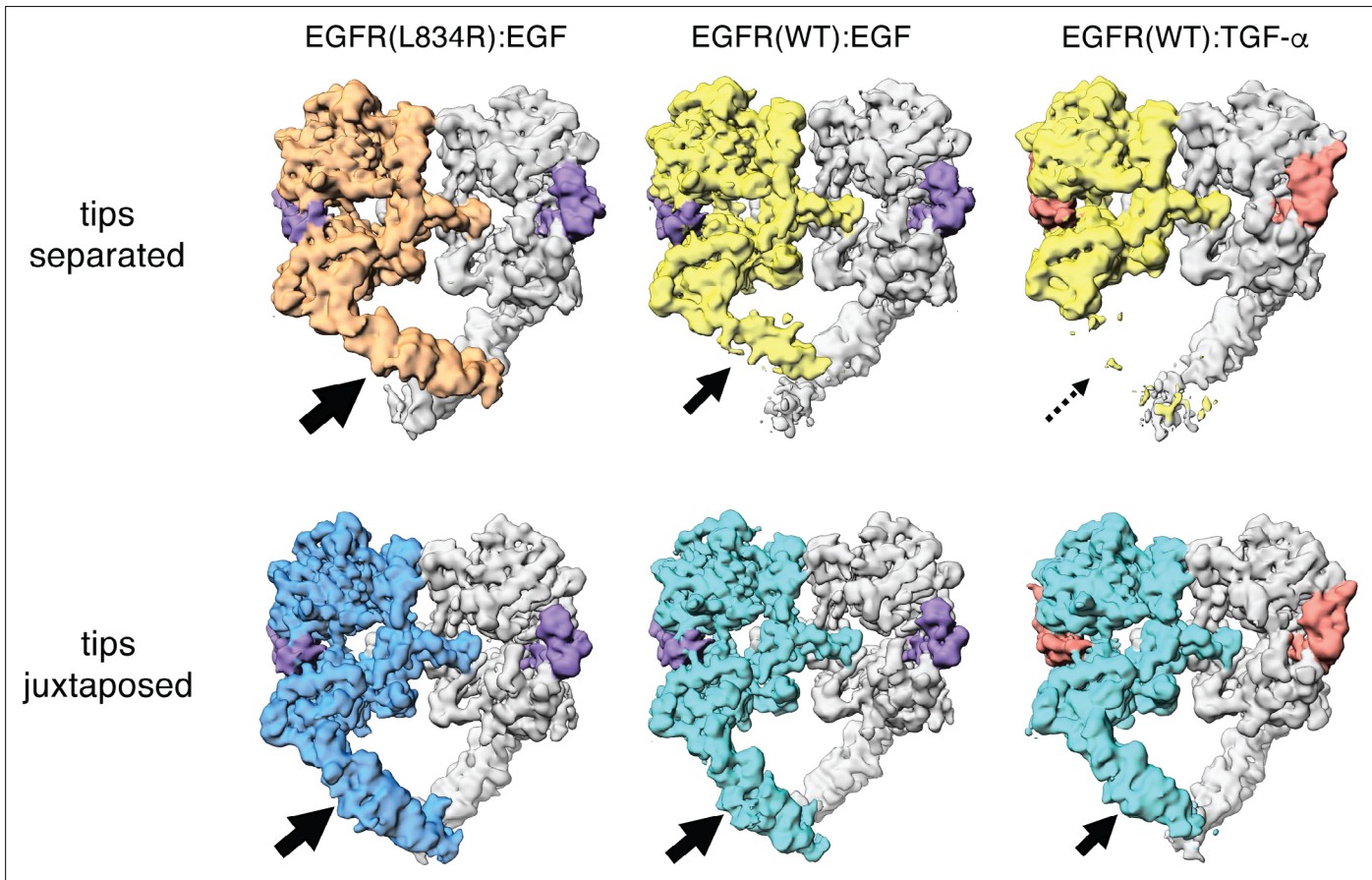

**Figure 7.** Comparison between Cryo-EM density maps of the tips-separated and the tips-juxtaposed conformations in different EGFR complexes. Cryo-EM density maps of the tips-separated (upper row) and the tips-juxtaposed (lower row) conformations in three different EGFR complexes are shown: EGFR(L834R) bound to EGF (left), EGFR(WT) bound to EGF (middle), and EGFR(WT) bound to TGF-α (right). The black arrows highlight the difference in stabilities of the flexible Domain IV legs in these complexes.

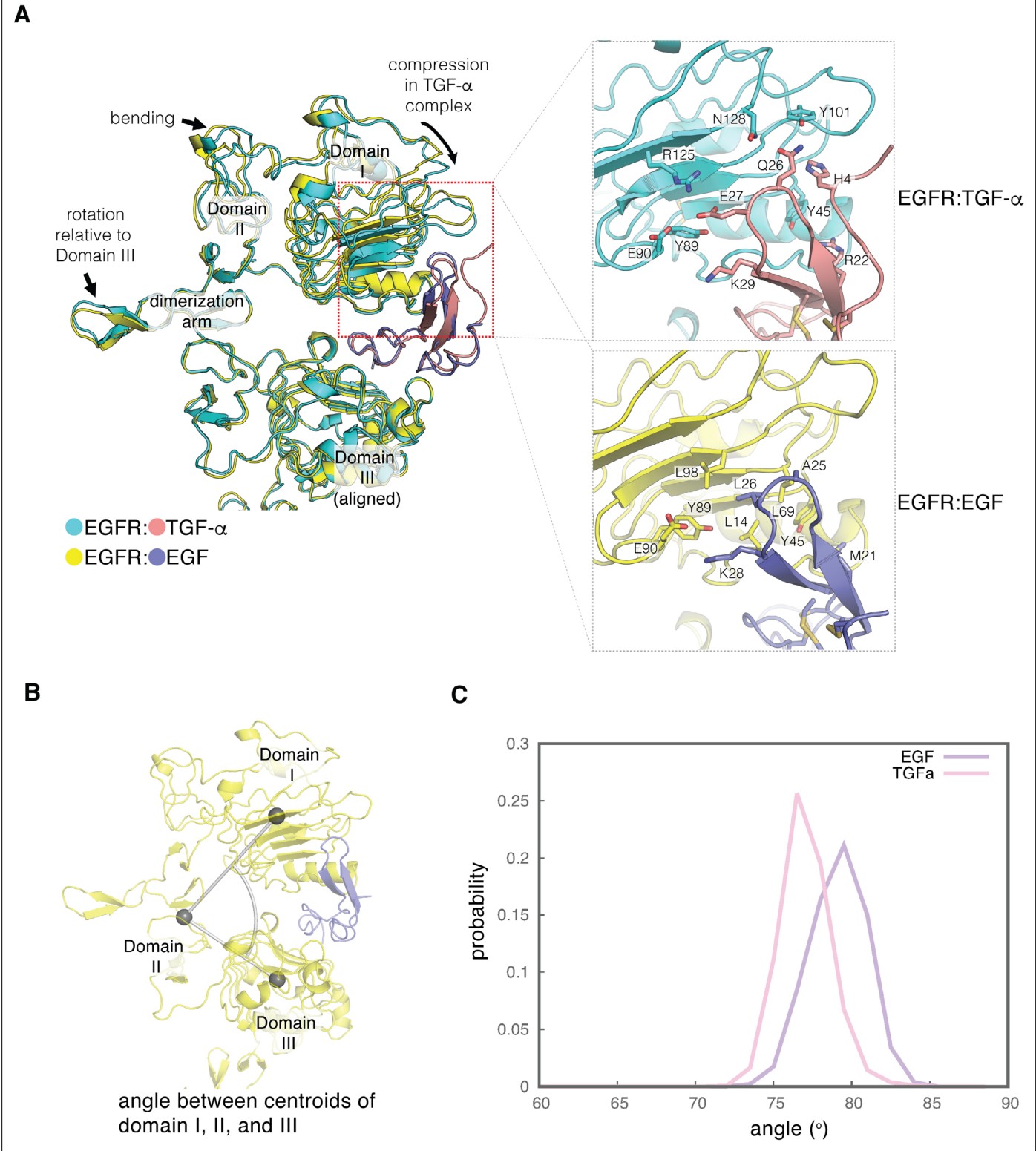

**Figure 8.** Difference in the ligand-binding heads between EGF- and TGF-α-bound EGFR complexes. (**A**) The differences in structure between the separated conformations of the EGF and TGF-α complexes of EGFR are illustrated. Structures corresponding to the separated conformations of the EGFR:EGF complex (yellow:slate) and the EGFR:TGF-α complex (cyan:salmon)are aligned on Domain III of each structure. Enlarged views of the interface between Domain I of EGFR and the ligands are shown on the right. The interface between Domain I of EGFR (cyan) and TGF-α (salmon) features hydrophilic residues that are shown (upper right). The equivalent interface between Domain I of EGFR (yellow) and EGF (slate) is characterized

*Figure 8 continued on next page*

*Figure 8 continued*

by hydrophobic residues (lower right). (**B**) The angle between centroids of Domain I (residues 7–200), part of Domain II (residues 260–285), and Domain III (317-470) is used to estimate the degree of closure of the ligand-binding site in a ligand-bound EGFR complex. (**C**), Probability distribution of the angles defined in **B**, sampled every nanosecond over ~1.8 μs and ~2.0 μs of MD simulations of the EGFR extracellular module bound to EGF (purple) and TGF-α (pink), respectively.

These results demonstrate that the structural difference between the two ligand-bound EGFR complexes can propagate from the ligand-binding head to the disposition of the Domain IV legs. Due to the more compressed ligand-binding pocket in the TGF-α complex, which leads to potential steric clashes between the Domain IV legs, TGF-α is less effective than EGF at stabilizing the Domain IV legs in the tips-separated conformation of the extracellular module.

## EGF and TGF-α induce different levels of phosphorylation in EGFR and Erk

Our cryo-EM analysis of the oncogenic EGFR variant (L834R) show that the stabilization of the flexible Domain IV leg in the separated conformation is correlated with the presence of an oncogenic mutation that results in increased levels of kinase activity. We speculate that the observed differences in the ability of EGF and TGF-α to stabilize the Domain IV leg in the tips-separated conformation underlies the ability of these two ligands to trigger different levels of kinase activity and downstream signaling from EGFR. Specifically, our structural model suggests that the EGF-bound EGFR complex is likely to be more active than the TGF-α bound EGFR.

Although previous studies have demonstrated that these two ligands can induce different long-term cellular responses (*Wilson et al., 2009*), it is unclear how this is manifested at the level of receptor autophosphorylation and the activation of proximal signaling events. To test the effect of these two ligands on the kinase activity of EGFR, we first purified detergent-solubilized full-length human EGFR from mammalian cells (HEK293FT), followed by stimulation of EGFR activity through addition of EGF and TGF-α, and detection of phosphorylation on EGFR by western blot analysis. Indeed, the results show that EGF induces stronger EGFR autophosphorylation than TGF-α (*Figure 9—figure supplement 1A*). Next, we transiently transfected mammalian cells with full-length human EGFR and stimulated the cells with saturating amounts (*Macdonald-Obermann and Pike, 2014*) of EGF and TGF-α (100 ng/ml), respectively. EGFR phosphorylation levels in the stimulated cells were analyzed by western blot. Consistent with the results obtained with the purified EGFR protein, the EGFR phosphorylation level is higher in the EGF-stimulated cells than that in the TGF-α-stimulated cells (*Figure 9—figure supplement 1B*). Importantly, ligand-dependent differences can also be detected in the downstream signaling events following receptor activation, by monitoring the phosphorylation level of the extracellular signal–regulated kinase (Erk). As shown in *Figure 9C* and *Figure 9—figure supplement 1C*, EGF stimulates higher levels of Erk phosphorylation than TGF-α does in cells transiently transfected with EGFR, at EGF and TGF-α concentrations of 100 ng/ml, 200 ng/ml, and 500 ng/ml.

According to our structural model, for these two ligands to differentially activate EGFR, the structural difference originating from the ligand-binding head needs to be transmitted to the tips of the extracellular module, and thereby to the transmembrane helices. We hypothesize that disrupting the conformational coupling between the ligand-binding head and the Domain IV leg, or the coupling between the Domain IV leg and the transmembrane helices, will interfere with the differential effects of EGF and TGF-α. To disrupt the connection between the ligand-binding head and the Domain IV leg, we mutated a conserved residue (Trp 492) buried in a hydrophobic pocket between Domain III and Domain IV (*Figure 9B*). Mutating this tryptophan to glycine (W492G) is expected to loosen the connection between Domain III and Domain IV, and the Erk phosphorylation assay shows that this mutation results in levels of Erk phosphorylation that are the same for activation by EGF and TGF-α (*Figure 9C*, *Figure 9—figure supplement 1E*).

Since our structural model suggests that the conformation of the linker at the junction between the extracellular module and the transmembrane helices is important for the coupling between the two domains, replacing the linker (Asn 615-Pro 620) with a flexible sequence, Gly-Gly-Ser-Gly-Gly-Ser (GGSGGS), should compromise this coupling. This mutation (replacement of the linker by GGSGGS) breaks the differential Erk activation by EGF and TGF-α (*Figure 9D and E*, *Figure 9—figure supplement 1E*). In another construct, we inserted a flexible segment, with sequence

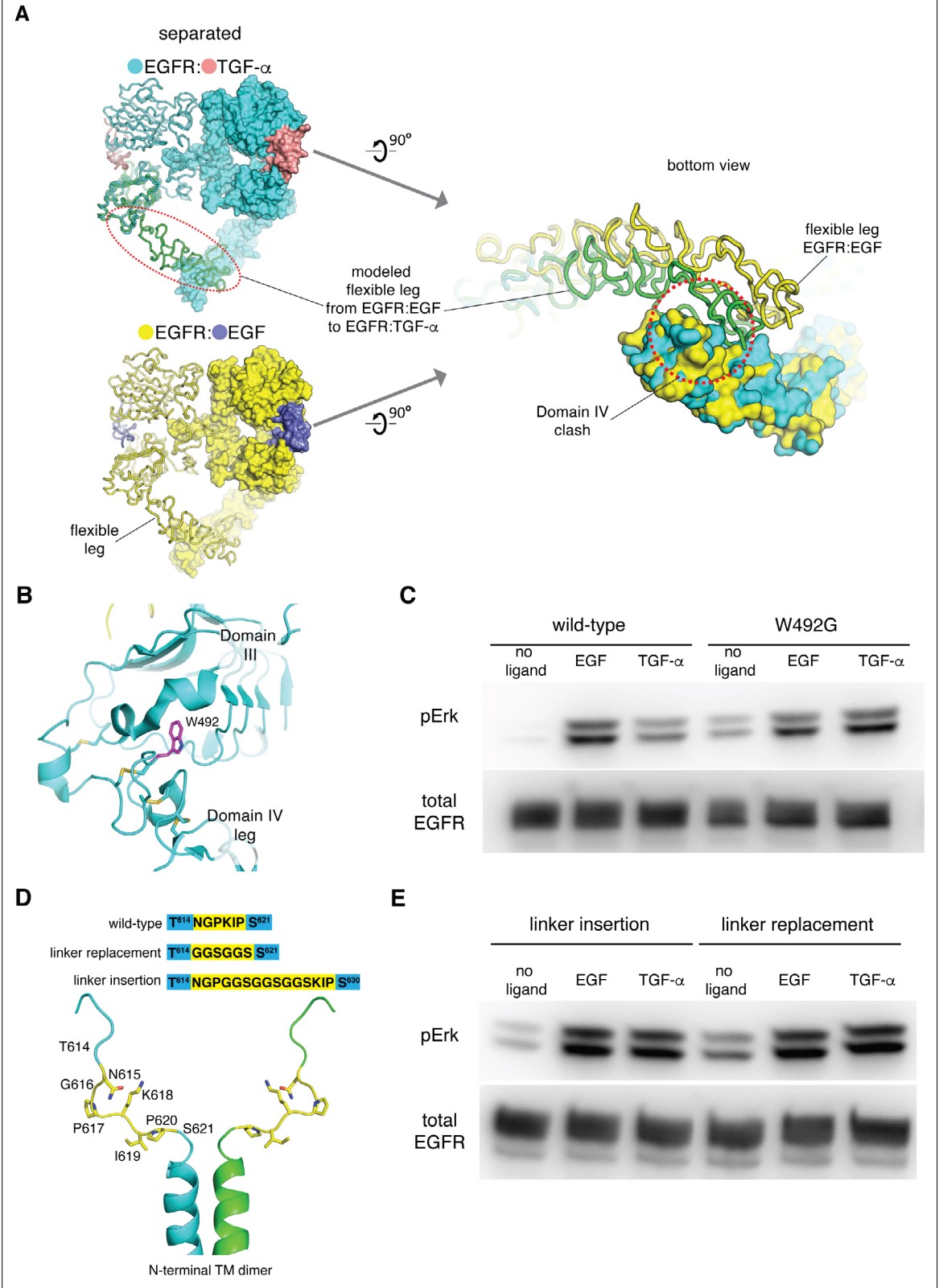

**Figure 9.** EGF and TGF-α differ in their abilities to stabilize the tips-separated conformation and to generate different levels of EGFR activation. (**A**) Destabilization of the flexible Domain IV leg in the separated conformation of the EGFR:TGF-α complex. The EGFR:TGF-α complex (upper left) is colored in cyan:salmon and the EGFR:EGFcomplex (lower left) is colored in yellow:slate. The flexible leg in the EGFR:TGF-α complex (green, red dotted line) is modeled based on the conformation of Domains III and IV in the left-hand subunit of the EGFR:EGF complex (lower left). When viewed from the

*Figure 9 continued on next page*

*Figure 9 continued*

bottom (right), the two Domain IV legs are accommodated in the EGFR:EGF complex (yellow), but not in the EGFR:TGF-α complex. There are steric clashes between the legs (green and cyan), within the dotted red line. (**B**) Location of the Trp 492 (purple), shown in the crystal structure of EGFR bound to EGF (cyan, PDB code: 3NJP). (**C**) Western-blot analysis of Erk phosphorylation in HEK293T cells transiently transfected with wild-type EGFR or EGFR variant, W492G. Anti-phospho-Erk antibody is used to detect phosphorylation of p44 and p42 MAP kinase (Erk1 and Erk2), and anti-FLAG antibody is used to detect the total amount of EGFR in the cell lysate loaded to each lane. (**D**) Mutations introduced to the linker at the junction between the extracellular module and the transmembrane helices, shown in the NMR structure of the N-terminal dimer configuration of the transmembrane helices (PDB code: 5LV6). (**E**) Western-blot analysis of Erk phosphorylation in HEK293T cells transiently transfected with two EGFR variants, linker insertion (left) and linker replacement (right). In linker insertion variant, Gly-Gly-Ser-Gly-Gly-Ser-Gly-Gly-Ser is inserted between Pro 617 and Lys 618. In linker replacement variant, the linker (Asn 615-Pro 620) is replaced by Gly-Gly-Ser-Gly-Gly-Ser.

The online version of this article includes the following figure supplement(s) for figure 9:

**Source data 1.** Original files of the full raw uncropped blots shown in *Figure 9*, with the relevant bands labeled.

**Source data 2.** Original files of the full raw uncropped blots shown in *Figure 9*, with the relevant bands labeled.

**Source data 3.** Original files of the full raw uncropped blots shown in *Figure 9*, with the relevant bands labeled.

**Source data 4.** Original files of the full raw uncropped blots shown in *Figure 9*, with the relevant bands labeled.

**Figure supplement 1.** EGF and TGF-α generate different levels of EGFR autophosphorylation and Erk phosphorylation.

**Figure supplement 1—source data 1.** Original files of the full raw uncropped blots shown in *Figure 9—figure supplement 1*, with the relevant bands labeled.

**Figure supplement 1—source data 2.** Original files of the full raw uncropped blots shown in *Figure 9—figure supplement 1*, with the relevant bands labeled.

**Figure supplement 1—source data 3.** Original files of the full raw uncropped blots shown in *Figure 9—figure supplement 1*, with the relevant bands labeled.

**Figure supplement 1—source data 4.** Original files of the full raw uncropped blots shown in *Figure 9—figure supplement 1*, with the relevant bands labeled.

Gly-Gly-Ser-Gly-Gly-Ser-Gly-Gly-Ser (GGSGGSGGS), into the linker connecting the extracellular module to the transmembrane helix, between Pro 617 and Lys 618. The insertion of this flexible segment results in a similar extent of Erk activation by EGF and TGF-α (*Figure 9D and E*, *Figure 9— figure supplement 1E*), which is consistent with the importance of this linker in coupling the conformations of the extracellular module and the transmembrane helix.

## Conclusions

Our cryo-EM analysis provides a molecular explanation for the ability of EGFR to respond differently to two high-affinity ligands, EGF and TGF-α (*Figure 10*). The differential activation of EGFR by these two ligands arises from structural differences in the receptor-ligand interactions. As a result, EGF and TGF-α differ in their abilities to stabilize the tips-separated conformation of the extracellular module that couples to the N-terminal dimer configuration of the transmembrane helices, which is further coupled to a specific configuration of the juxtamembrane segments (*Arkhipov et al., 2013*; *Endres et al., 2013*; *Jura et al., 2009*; *Sinclair et al., 2018*), ultimately promoting the formation of kinase dimers with higher activity (*Zhang et al., 2006*). In this way, EGF and TGF-α can generate different responses from EGFR by differentially modulating the structure of the junction point between the extracellular module and the transmembrane helices, altering the balance between alternative dimeric configurations of the transmembrane helices (*Figure 10B*). A lack of rigidity in the linkage between the extracellular module and the transmembrane helices is emerging as a common feature in receptor tyrosine kinases (*Diwanji et al., 2021*; *Li et al., 2019*; *Lu et al., 2010*; *Uchikawa et al., 2019*; *Zhang et al., 2020*), and the mechanism we have defined, involving modulation of the separation of the tips of the extracellular modules, could potentially be utilized by other receptor tyrosine kinases to generate ligand-specific signaling outputs (*Ahmed et al., 2020*).

Members of the EGFR family are important targets for the development of cancer therapeutics, including antibodies that block receptor dimerization and activation (*Arteaga and Engelman, 2014*; *Cai et al., 2020*). Our structural analysis suggests new strategies for the modulation of EGFR signaling. For example, antibodies or peptides that can bind to the Domain IV legs and stabilize the juxtaposed conformation would allow the receptor to dimerize but send weaker signals. Such a strategy may

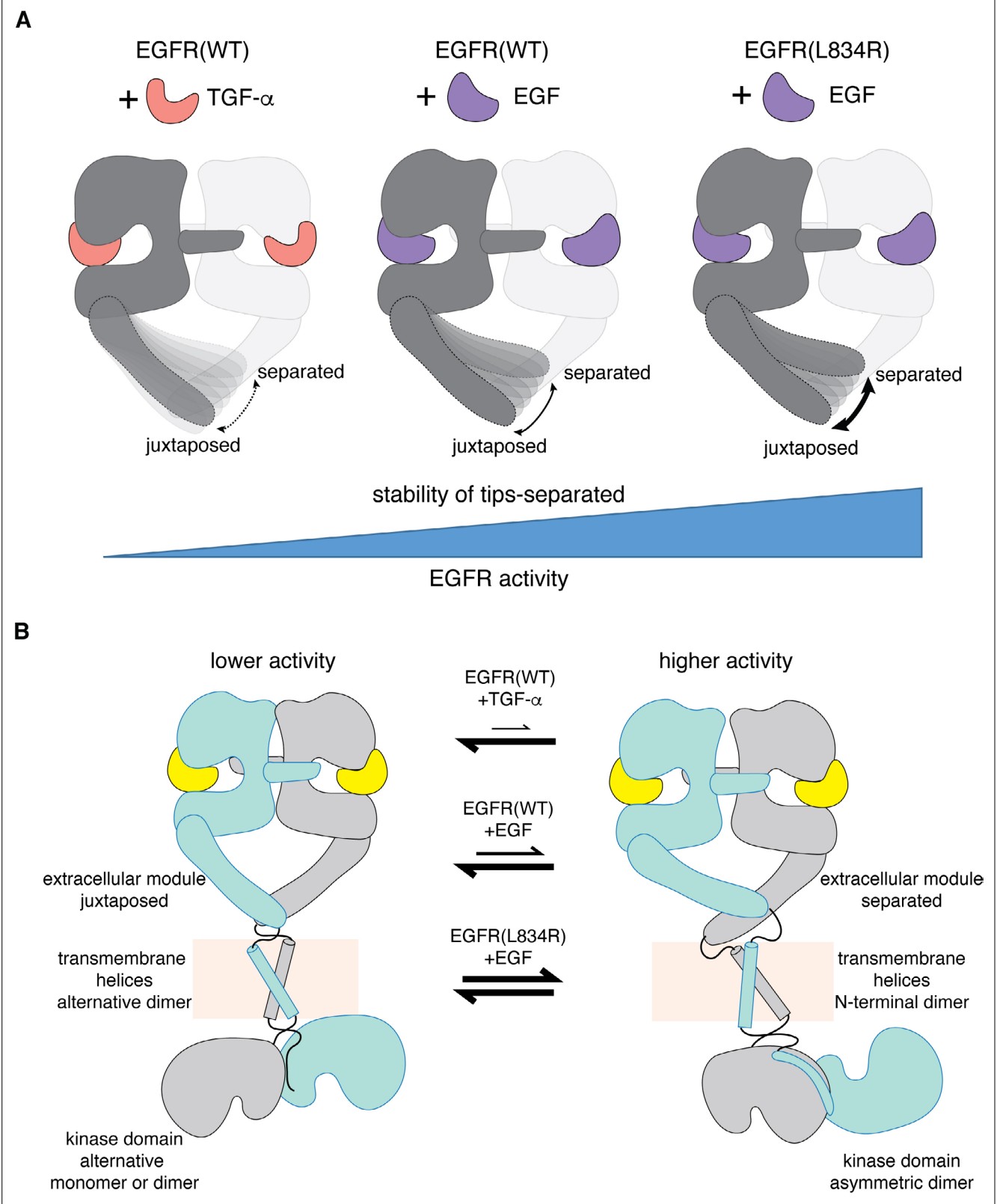

**Figure 10.** A mechanism for differential activation of EGFR by EGF and TGF-α. (**A**) Schematic representations of the ligand-bound extracellular module of EGFR illustrate the differential modulation of the conformation of the Domain IV legs by TGF-α (left), EGF(middle) and oncogenic L834R mutation (right). Cryo-EM densities for the Domain IV legs in the juxtaposed conformation is similarly defined in all three complexes (EGFR(WT):TGF-α, EGFR(WT):EGF, and EGFR(L834R):EGF). In contrast, density for the Domain IV legs in the separated conformation is poorly defined in the

*Figure 10 continued on next page*

*Figure 10 continued*

EGFR(WT):TGF-α complex and best defined in the EGFR(L834R):EGF complex. Increased stability of the tips-separated conformation of the extracellular module is correlated with higher level of EGFR activity. (**B**) Schematic diagrams illustrate two extreme dimeric conformations of the ligands-bound full-length EGFR, representing a lower activity state (left) and a higher activity state (right). It shows how different EGFR ligands and oncogenic mutation could generate different EGFR activity by altering the balance between these dimeric conformations of the full-length EGFR.

circumvent resistance mechanisms that arise in response to antibodies and inhibitors that shut down EGFR signaling completely (*Brand et al., 2011*; *Chong and Jänne, 2013*; *Cai et al., 2020*).

## Materials and methods
### Cell culture
#### Insect cells
*Spodoptera frugiperda* Sf9 cells were maintained as suspension cultures at 27 °C in ESF-921 Insect Cell Culture Medium (Expression Systems). Isolation of bacmid DNA, transfection of Sf9 cells, and amplification of baculovirus were performed following the protocols detailed in the Bac-to-Bac Baculovirus Expression System (ThermoFisher Scientific).

#### Mammalian cells
Human HEK293S GnTI⁻ cells were cultured as suspension at 37 °C in FreeStyle293 expression medium (Invitrogen) supplemented with 2 % (vol/vol) FBS, in the presence of 8 % $CO_2$. The HEK293S GnTI⁻ suspension cells were maintained between 0.2 and $3 \times 10^6$ cells/ml. HEK293T and HEK293FT cells were cultured at 37 °C in complete Dulbecco's modified Eagle's medium (DMEM) supplemented with GlutaMAX (ThermoFisher) and 10 % (vol/vol) FBS, in the presence of 5 % $CO_2$.

#### Cell line sources
Human HEK293T (ATCC CRL-3216), Human HEK293S GnTI- (ATCC CRL-3022), Spodoptera frugiperda Sf9 (ATCC CRL-1711). All the cell lines are authenticated and tested by manufacturer for mycoplasma contamination. No additional test was performed by the authors of this study. No commonly misidentified lines were used.

### Full-length EGFR protein expression and purification for Cryo-EM studies
#### Protein expression
A cDNA fragment encoding full-length human EGFR (UniProt accession no. P00533) was cloned into the pEG_BacMam vector (*Goehring et al., 2014*). To facilitate affinity purification, a FLAG-tag (with sequence DYKDDDDK) was inserted after the signal sequence of EGFR. To monitor protein expression and to reduce non-specific degradation of the C-terminal tail of EGFR, EGFP was fused after the C-terminus of EGFR. An oncogenic EGFR variant (L834R) was generated by site-directed mutagenesis.

Plasmid DNA carrying the EGFR sequence was transformed into DH10Bac competent cells (ThermoFisher Scientific) to produce the bacmid DNA, following the protocol detailed in the Bac-to-Bac Baculovirus Expression System. To produce the recombinant baculovirus, purified bacmid DNA was transfected into Sf9 cells using Cellfectin II (ThermoFisher Scientific). After two rounds of amplification, the recombinant baculovirus was used to infect HEK293S GnTI⁻ cells at a 1:20 (vol/vol) ratio. After 24 hr, the infected HEK293S GnTI⁻ suspension culture was supplemented with 2 μM of Erlotinib, an EGFR inhibitor, and 8 mM sodium butyrate, for optimal protein expression and stability. Cells were harvested by centrifugation 72 hr after the infection.

#### Protein purification
Harvested cells were resuspended in lysis buffer (25 mM Tris pH 8.0, 150 mM NaCl, 2 μM Erlotinib and cOmplete Protease Inhibitor Cocktail (Roche)) and were lysed by sonication in pulse mode for 10 min, with 2 s intervals between each 1 s pulse. Large debris and unbroken cells were removed from the crude lysate by centrifugation for 12 min at 19,000 g. The membrane fraction was pelleted by ultra-centrifugation at 190,000 g for 1 hr and resuspended in lysis buffer. To extract EGFR from the

membrane, 1 % DDM (Dodecyl-β-D-Maltoside, Anatrace) and 0.1 % CHS (Cholesteryl Hemisuccinate, Anatrace) were added to the suspended membrane fraction. After mixing for 2 hr at 4 °C, detergent solubilized proteins were collected by ultra-centrifugation at 190,000 g for 30 min. The supernatant was incubated with Anti-FLAG M2 Affinity agarose gel (Sigma-Aldrich) for 4 hr at 4 °C. The gel was washed with lysis buffer supplemented with 0.05 % DDM and 0.005 % CHS. EGFR was eluted from the gel by lysis buffer containing 0.16 mg/ml FLAG peptide, 0.05 % DDM and 0.005 % CHS.

Affinity-purified full-length EGFR was concentrated to 50 µM and treated with 100 µM human EGF (Novoprotein) or TGF-α (Sino Biological) for at least 5 min on ice, before further fractionation by size-exclusion chromatography using a Superose 6 Increase 10/300 GL column (GE Healthcare), with buffer containing 25 mM Tris pH 8.0, 150 mM NaCl, 2 µM Erlotinib, 0.02 % DDM and 0.002 % (CHS). The peak fractions containing ligand-bound dimeric EGFR were collected and immediately used for reconstitution and the reconstituted receptor was applied to EM grids. The fractions were also flash-frozen in liquid nitrogen and used for reconstitution at later times. Freezing and thawing the fractions more than once leads to deterioration in cryo-EM sample quality.

## Reconstitution of ligand-bound full-length EGFR complex

### Reconstitution into lipid nanodiscs

Purified protein complexes of EGFR bound to EGF or TGF-α were reconstituted into lipid nanodiscs as previously described, with modifications (*Ritchie et al., 2009*). Briefly, membrane scaffold proteins MSP1D1 and MSP1E3D1 were expressed and purified from *Escherichia coli*. 10 mg soybean PC (L-α-phosphatidylcholine (soy), Avanti) dissolved in chloroform was dried using nitrogen stream and residual chloroform was further removed by desiccation under vacuum overnight. Rehydration buffer containing 25 mM Tris pH 8.0, 150 mM NaCl, 0.2 mM DTT, 1 % DDM and 1 % sodium cholate was added to the dried lipid and vortexed briefly, resulting in a semi-clear solution at 10 mM lipid concentration. After three cycles of freezing (in liquid nitrogen)-and-thawing (warm water bath), the lipid solution was bath sonicated for 1 hr. The resulting clear lipid stock (10 mM) was immediately used for lipid nanodiscs assembly or stored at –80 °C in small aliquots. Purified complexes of EGFR bound to EGF or TGF-α (~10 µM, solubilized in 0.02 % DDM and 0.002 % CHS) was supplemented with 20 µM extra EGF or TGF-α to keep the ligand-binding site on the receptor saturated, before it was mixed with the soybean PC stock and membrane scaffold proteins at molar ratios of 1:6:240 (EGFR monomer:MSP1D1:soy PC) and 1:6:600 (EGFR monomer:MSP1E3D1:soy PC). The reaction mixture (~500 µl) was incubated on ice for 30 mins, before 20 mg Bio-beads SM2 absorbent (Bio-Rad) were added to initiate the removal of detergents from the mixture and to start the nanodiscs assembly process. The mixture was incubated at 4 °C by continuous rotation. After 1 hr, a second batch of Bio-beads (20 mg) was added to the mixture to replace the first batch, and the sample was incubated for overnight at 4 °C. A third and final batch of Bio-beads (20 mg) was added to replace the second batch, 2 hrs before the reconstitution mixture was cleared by removing the Bio-beads through centrifugation. After another centrifugation at 14,000 rpm for 10 min, the reconstitution mixture was fractionated by size-exclusion chromatograph using Superose 6 Increase 10/300 GL column (GE Healthcare) in detergent-free buffer containing 25 mM Tris pH 8.0, 150 mM NaCl and 5 µM Erlotinib. The peak fraction corresponding to dimeric EGFR:ligands complexes reconstituted in lipid nanodisc was collected and analyzed by SDS-PAGE, before being applied to negative-stain and cryo-EM grids.

### Reconstitution into amphipols

Samples containing purified complexes of EGFR bound to EGF or TGF-α (~10 µM, solubilized in 0.02 % DDM and 0.002 % CHS) were supplemented with 20 µM extra EGF or TGF-α, before the amphipol A8-35 (Anatrace) was added to reach a mass ratio of 1:3 (EGFR:A8-35). After incubation at 4 °C overnight, free amphipol molecules were removed by size-exclusion chromatography using a Superose 6 Increase 10/300 GL column in detergent-free buffer containing 25 mM Tris pH 8.0, 150 mM NaCl and 5 µM Erlotinib. Peak fractions corresponding to the reconstituted EGFR:ligand complexes in amphipol were analyzed by SDS-PAGE, before EM studies.

### Reconstitution into peptidiscs

Samples containing purified complexes of EGFR bound to EGF or TGF-α (~10 µM, solubilized in 0.02 % DDM and 0.002 % CHS) were supplemented with 20 µM extra EGF or TGF-α, before the

addition of $NSP_r$ peptide (*Carlson et al., 2018*) to the samples at a mass ratio of 1:1.5 (EGFR:$NSP_r$ peptide). After overnight incubation at 4 °C, the sample was fractionated using a Superose 6 Increase 10/300 GL column (detergent-free buffer: 25 mM Tris pH 8.0, 150 mM NaCl and 5 µM Erlotinib) and free $NSP_r$ peptides were removed during fractionation. Peak fractions corresponding to the reconstituted EGFR:ligand complexes in peptidiscs were analyzed by SDS-PAGE, before applied to EM grids.

## Preparation of electron microscopy (EM) grids and data acquisition

For negative-stain EM, a 5 µL sample of reconstituted full-length human EGFR (~0.05 mg/mL protein in 25 mM Tris pH 8.0, 150 mM NaCl and 2 µM Erlotinib) was placed on the continuous-carbon side of a glow-discharged copper grid (Ted Pella, Redding, CA, USA), and the excess sample was removed by wicking with filter paper after 1 min incubation. The bound particles were stained by floating the grids on four consecutive 30 µL droplets of freshly prepped 2 % uranyl formate solution and incubating the grids with each droplet for 15 s. The excess stain was removed by blotting with filter paper and the grids were air-dried for at least 5 min. Images of negative-stained full-length human EGFR reconstituted in lipid nanodisc, amphipol, and peptidisc were recorded on a 4049 × 4096 pixels CMOS camera (TVIPS TemCam-F416) using the automated Leginon data collection software (*Suloway et al., 2005*). Samples were imaged using a Tecnai 12 transmission electron microscope (FEI, Hillsboro, OR, USA) at 120 keV with a nominal magnification of 49,000 x (2.18 Å calibrated pixel size at the specimen level) using a defocus range of –0.8 to –1.5 µm. All data were acquired under low dose conditions, allowing a dose at around 35 $e^-$/Å$^2$.

For cryo-EM, UltrAuFoil R 1.2/1.3 300 mesh grids (Electron Microscopy Sciences, Q350AR13A) were rendered hydrophilic by pretreatment with a $O_2$/$H_2$ gas mixture using a Solarus 950 plasma cleaning system (Gatan, Inc, Pleasanton, CA) for 15 s at 50 W. After application of 3 µL sample of reconstituted full-length human EGFR (~0.5 mg/ml protein in 25 mM Tris pH 8.0, 150 mM NaCl and 2 uM Erlotinib), grids were plunge-frozen using a Leica EM GP (Leica Microsystems Inc, Buffalo Grove, IL) with blotting time of 8 s, chamber temperature and humidity of +4 °C and 95%, and a liquid ethane temperature of –184 °C. The grids were imaged using a Titan Krios transmission electron microscope (Thermo Fisher Scientific, Waltham, MA) operated at 300 kV. Images were recorded on a K3 camera (Gatan, Inc) operated in super-resolution counting mode with a physical pixel size of 0.54 Å. The detector was placed at the end of a GIF Quantum energy filter (Gatan, Inc), operated in zero-energy-loss mode with a slit width of 20 eV. Due to strongly preferred particle orientations, all movies were collected at 40° tilt, at a nominal magnification of 81,000 x and a defocus range between –0.75 and –1.5 µm., using the automated data collection software Latitude (Gatan, Inc). The nominal dose per frame was ~1 $e^-$/Å$^2$ and the total dose per movie is ~50 $e^-$/Å$^2$. For the full-length EGFR:EGF complex reconstituted in peptidiscs, a total of 4331 movies were collected. For the full-length EGFR:TGF-α complex reconstituted in peptidiscs, a total of 7406 movies were collected. For the full-length EGFR(L-834R):EGF complex reconstituted in peptidiscs, a total of 5013 movies were collected.

## EM image processing

For negative-stain EM images of full-length EGFR:EGF complexes reconstituted in lipidic nanodiscs (MSP1E3D1), the initial image processing and classification steps were performed using the Appion image processing environment (*Lander et al., 2009*). Particles were first selected from micrographs using DoG Picker (*Voss et al., 2009*). The contrast transfer functions (CTFs) of the micrographs were estimated using the CTFFIND3 (*Mindell and Grigorieff, 2003*). CTF correction of the micrographs was performed by Wiener filter using ACE2 (*Mallick et al., 2005*). A total of 112,408 particles were extracted using a 192 × 192 box size and binned by a factor of 2. Each particle was normalized to remove pixels whose values were above or below 4.5 σ of the mean pixel value using the XMIPP normalization program (*Scheres et al., 2008*). In order to remove incorrectly selected protein aggregates or other artifacts, particles with extreme intensity values were removed. The remaining 84,260 particles were subjected to 2D iterative reference-free alignment and classification using IMAGIC multi-reference alignment (MRA) and a topology-representing network classification (*van Heel et al., 1996*; *Ogura et al., 2003*), to produce the 200 2D class averages.

For Cryo-EM images of full-length EGFR:EGF complex reconstituted in peptidiscs, Extended Data *Figure 3* shows a summary of the image processing workflow. In brief, movies were motion-corrected using the MotionCor2 wrapper (*Zheng et al., 2017*) in Relion3 (*Zivanov et al., 2018*).

Corrected micrographs were imported to cryoSPARC v2 (*Punjani and Fleet, 2020*; *Punjani et al., 2017*) for further processing. After local CTF determination using Patch-Based CTF Estimation, a total of 4,234,635 particles from 4,431 micrographs were automatically selected using template-free Blob_picker and were extracted with a box size of 320 × 320 pixels. Due to strongly preferred particle orientations, a relatively low threshold value was used during the particle picking to include particles with less contrast than the preferred views. The majority of the initial 4,234,635 picks are from the background. After iterative 2D classification, most of the background particles were removed based on 2D class averages. The remaining 1,587,872 particles were further cleaned-up by ab-initio reconstruction into four classes. The resulting 3D maps showed that 3 of the ab-initio reconstruction classes were mostly 'junk' particles without recognizable structural features, and the other class showed clear resemblance to the dimeric extracellular module of EGFR. This clean class of particles (814,089) were selected and subjected to a final round of 2D classification. After visual inspection of the final 2D class averages, 765,883 particles were retained and ab-initio reconstruction was performed to generate the initial model, which was used as the reference for a round of homogeneous refinement as well as a non-uniform refinement (*Punjani et al., 2019*). Based on the gold-standard Fourier shell correlation (FSC) 0.143 criterion, the overall resolution of the reconstructions from homogeneous refinement and non-uniform refinement are determined as 3.0 Å and 2.9 Å, respectively. After visual inspection of both reconstructions, the map generated from non-uniform refinement with better quality was chosen for further analysis and used as the reference in the subsequent round of heterogeneous refinement (3D classification) using the 765,883 particles, resulting in 10 classes. A final round of homogeneous refinement and non-uniform refinement were performed for each of the resulting 10 classes from the heterogeneous refinement run. After visual inspection, the 10 maps resulting from non-uniform refinement were chosen for model building and refinement for each class. The overall resolution of these 10 maps ranges between 3.1 Å and 3.7 Å.

Cryo-EM data processing of full-length EGFR(L834R):EGF complex and full-length EGFR:TGF-α complex reconstituted in peptidiscs followed similar procedures described above. For full-length EGFR(L834R):EGF complex, after local CTF estimation by Patch-Based CTF Estimation, a total of 3,250,237 particles were automatically picked by Blob_Picker. After iterative 2D classification and ab-initio reconstruction, background and spurious particles were removed. 987,367 cleaned particles were subjected to homogeneous as well as non-uniform refinement, using the initial model generated from ab initio reconstruction as reference. After visual inspection, the map generated by non-uniform refinement (overall resolution: 3.0 Å) was chosen as the reference for the subsequent heterogeneous refinement, resulting in 10 classes. A final round of non-uniform refinement was performed for each of the 10 classes and the resulting 10 maps were used for model building and refinement. The resolutions of these 10 maps were determined to be in the range of 3.2–4.2 Å.

For the full-length EGFR:TGF-α complex, a total of 6,846,834 particles were initially picked by Blob_Picker. Iterative 2D classification and ab-initio reconstruction were performed to remove background and spurious particles, and 1,019,024 cleaned particles were retained. These cleaned particles were subjected to homogeneous refinement and non-uniform refinement. The map resulting from non-uniform refinement was at an overall resolution of 3.2 Å and it was used as the reference for the subsequent heterogeneous refinement, resulting in 10 classes. A final round of non-uniform refinement was performed for each of the 10 classes and the resulting 10 maps were determined at resolutions between 3.4 Å and 3.9 Å. These maps were used for model building and refinement.

3D variability analysis, local resolution estimation, and B-factor sharpening of all refined maps were performed using the cryoSPARC v2.

## Model building and refinement

To generate models for different conformations of the EGFR(WT):EGF and EGFR(L834R):EGF complexes, a crystal structure of the isolated EGFR extracellular module bound to EGF (PDB code: 3NJP) was split into four fragments: two heads and two legs. The two heads contain EGFR residues 1–500 and EGF residues 5–51 from each side of the dimeric structure, respectively. The two legs contain EGFR residues 501–614 from each EGFR molecule, respectively. These four fragments were fit into the reconstructed cryo-EM maps of the EGFR(WT):EGF and EGFR(L834R):EGF complexes, using UCSF Chimera (*Pettersen et al., 2004*). UCSF chimera-fitted models were subjected to one round of real-space refinement (*Afonine et al., 2018*) in *Phenix* (*Liebschner et al., 2019*). To prevent overfitting

due to the lower local resolution in the map region corresponding to the flexible Domain IV legs, the coordinates of the flexible Domain IV leg was not refined against the cryo-EM maps. The models were manually adjusted in Coot (*Emsley et al., 2010*), followed by iterative rounds of real-space refinement in Phenix and manual fitting in Coot. Model quality was assessed using the comprehensive model validation tools from Phenix (*Figure 3—source data 1*). All maps and models were visualized using UCSF Chimera and PyMOL (The PyMOL Molecular Graphics System, Version 2.0 Schrödinger, LLC).

Similar procedures were followed to generate models for the different conformations of the EGFR:TGF-α complex, except that a crystal structure of the truncated EGFR extracellular domains bound to TGF-α (PDB code: 1MOX) was used to generate the fragments of the heads for the initial model fitting in UCSF Chimera.

## Molecular dynamics simulations

### System preparation for molecular dynamics simulations of extracellular module

We built a system comprising of the extracellular module bound to EGF starting from the crystal structure (PDB code: 3NJP *Lu et al., 2010*). The N-terminal ends of each EGFR subunit are the true termini and were left uncapped. The C-terminal ends of EGFR were capped with a methyl group. In the crystal structure, four residues are missing from the N-terminal ends of the either ligand, and two residues are missing from the C-terminal ends. We initiated one trajectory starting with these structures while capping the N-terminal ends with acetyl groups and the C-terminal ends with methyl groups. For three other trajectories, we built in the missing residues with PyMOL (*Schrödinger, LLC, 2015*) and left the termini uncapped in these structures. These systems were solvated such that the water layer extended for 15 Å beyond the longest dimension of the protein, and ions were added such that the ionic strength was 0.15 M with VMD (*Humphrey et al., 1996*).

We built a second system of EGFR bound to TGFα. For this structure, EGFR was taken from the crystal structure 3NJP (*Lu et al., 2010*), and TGFα was taken from the structure of truncated EGFR bound to TGFα (PDB code: 1MOX *Garrett et al., 2002*). Similar to the system of EGFR bound to EGF, the EGFR subunits were left uncapped at the N-terminal ends and capped at the C-terminal ends. The ligands were uncapped in all the simulations. These systems were solvated such that the water layer extended for 15 Å beyond the longest dimension of the protein and ions were added such that the ionic strength was 0.15 M with VMD (*Humphrey et al., 1996*).

## Minimization, equilibration, and production protocols for simulations of extracellular module

The energies of each system were minimized, first for a 1000 steps while holding protein atoms fixed, and then for a 1000 steps while allowing all the atoms to move. The systems were then equilibrated, first for one ns while holding all the heavy atoms fixed, then for one ns while holding the protein backbone atoms fixed, and then for one ns without any constraints. The four trajectories for each system were then allowed to run for between ~350 and ~ 550 ns, for a total simulation time of ~1.8 μs for EGFR bound to EGF and ~2.0 μs for EGFR bound to TGF-α. All equilibration and production simulations were carried out at a temperature of 300 K and pressure of 1 atm.

## System preparation for molecular dynamics simulations of transmembrane helices in POPC membrane

We built two systems – the first of the two transmembrane helices (residues 610–653) associating at the N-terminal ends starting from model 1 of the NMR structure (PDB code: 5LV6 *Bocharov et al., 2017*), and the second of the two transmembrane helices (residues 614–653) associating at the C-terminal ends starting from model 6 of the NMR structure (PDB code: 2M0B *Bocharov et al., 2016*). We used the Membrane Builder module on the CHARMM-GUI website (*Jo et al., 2008*; *Wu et al., 2014*) to insert each protein system into a lipid bilayer comprising of neutral POPC lipids. The x- and y- dimensions of the bilayer was set to 100 Å each. This lipid bilayer comprised of 148 lipids in the upper leaflet, and 146 lipids in the lower leaflet in the N-terminal dimer system, and 142 lipids in each leaflet in the C-terminal dimer system. The N-termini of each protein chain were capped with acetyl groups and the C-termini were capped with methyl groups. The system was solvated such that the solvent layer extended to 20 Å beyond the lipid bilayer on either side, and ions were added such that

the ionic strength was 0.1 M, using VMD (*Humphrey et al., 1996*). The final edge dimensions of these systems were ~103 Å x 103 Å x 118 Å.

## Minimization, equilibration, and production protocols

For each system, first the energy was minimized while holding the protein, waters, ions, and lipid heads fixed, and allowing only the lipid tails to move, for 1000 steps, then while holding the protein and lipid heads fixed for 5000 steps, then while holding only the protein fixed for 1,000 steps, and finally for a 1000 steps while allowing all the atoms to move. This system was then equilibrated using molecular dynamics for five ns while holding the protein fixed, to allow the lipids to relax about the protein.

For the N-terminal dimer system, a longer trajectory of 175 ns then was generated from this equilibrated structure, during which all the atoms were allowed to move. During the course of this simulation, the N-terminal ends of the linkers (residues 610–613), which are disordered in this structure, interacted frequently with each other. In the full-length structure, these residues are a part of the ordered region in Domain IV of the extracellular module and would not be available to form such interactions. We therefore deleted these four residues, such that the protein chains started at Thr 614, and capped these N-terminal ends with acetyl groups. The energy of this system was minimized with the same multi-step protocol as before and used to initiate 25 independent molecular dynamics trajectories.

For the simulations starting from the C-terminal dimer, in the initial structure the gap between the N-terminal ends is too small to accommodate lipid tails. Additionally, the helices tilt away from each other such that the distance between the Thr 627 residues at the N-terminal ends of the helices changes from ~18 Å to ~23 Å, in order to optimally pack the hydrophobic residues of the protein into the membrane (*Killian and Nyholm, 2006*). In order the maintain the N-terminal ends of the helices, a multi-step equilibration protocol was used. First, the distance between the Thr 627 residues of either subunit was increased from ~18 Å to ~23 Å over five ns. During this phase, restraints applied to the protein backbone atoms of residues 620–626 to maintain the Ile619-Ala623 association and the $3_{10}$-helical turn at the N-terminal ends of the helices. The system was equilibrated for another 25 ns with these restraints applied. The system was then run for 20 ns without restraints, to ensure that the N-terminal ends were stable. This structure was then used to start 25 productions runs which ran for 100ns each for a total of 2.5 μs. All equilibration and production simulations were carried out at a temperature of 300 K and pressure of 1 atm.

## Simulation parameters

The NAMD package was used to run the minimization and equilibration simulations (*Phillips et al., 2005*) with the CHARMM36m force field (*Huang et al., 2017*). The velocity Verlet algorithm was used to calculate the trajectories of the atoms. A time step of 2 fs was used. Particle Mesh Ewald was used to calculate long-range electrostatic interactions, with a maximum space of 1 Å between grid points (*Darden et al., 1993*). Long-range electrostatics were updated at every time step. Van der Waals interactions were truncated at 12 Å. Hydrogen atoms bonded to the heavy atoms were constrained using the ShakeH algorithm (*Ryckaert et al., 1977*). The temperature was controlled with Langevin dynamics with a damping coefficient of 1/ps, applied only to non-hydrogen atoms. Pressure was controlled by the Nose-Hoover method with the Langevin piston, with an oscillation period of 200 fs and a damping time scale of 50 fs (*Feller et al., 1995*; *Martyna et al., 1994*).

Distance and angle variations over the course of the simulations were calculated with the Amber-Tools20 package (*Case et al., 2020*).

## Free energy calculations with umbrella sampling

Umbrella sampling simulations (*Torrie and Valleau, 1977*) were carried out to measure the free energy change as the distance between the Thr614 residues was decreased from 27 Å to 8 Å when the transmembrane helices are in the N-terminal dimer conformation. We performed three umbrella sampling calculations, starting from initial paths generated from three different steered molecular dynamics simulations. For each umbrella sampling calculation, 20 windows of simulations were run, where each window was separated by 1 Å, and the starting structure in each window had distances between Thr614 distances ranging from 8 Å to 27 Å. A force constant of 1 kcal.mol⁻¹.Å⁻² was used to maintain

the distance between Thr 614 residues in each window, using the colvars module (*Fiorin et al., 2013*). Two of the umbrella sampling simulations converged over 60 ns while the third converged over 90 ns. The biased probability distribution thus generated was reweighted and the free energy change was calculated using the Weighted Histogram Analysis Method (*Kumar et al., 1992*) using the implementation provided by Alan Grossfield (*Grossfield, 2010*).

## Western-blot analysis of EGFR autophosphorylation with purified EGFR proteins

### Expression and purification of FLAG-tagged full-length human EGFR from HEK293FT cells

A cDNA fragment encoding full-length human EGFR (UniProt accession no. P00533) was cloned into the pEGFP-N1 plasmid (Clontech, Mountain View, CA). To facilitate affinity purification, a FLAG-tag (with sequence DYKDDDDK) was inserted after the signal sequence of EGFR. To monitor protein expression and to reduce non-specific degradation of the C-terminal tail of EGFR, EGFP was fused after the C-terminus of EGFR. HEK293FT cells were transfected with the plasmid using FuGENE6 (Promega). After 24 hr, 10 mM sodium butyrate was added to the culture medium. The cells were harvested from culture dishes 36 hr after transfection and were resuspended in lysis buffer (25 mM Tris pH 8.0, 150 mM NaCl, and cOmplete Protease Inhibitor Cocktail). To extract EGFR from the membrane, 1 % DDM and 0.1 % CHS were added to the cell suspension. After mixing for 1 hr at 4 °C, detergent solubilized proteins were collected by ultra-centrifugation at 190,000 g for 40 min. The supernatant was incubated with Anti-FLAG M2 Affinity agarose resin for at least 4 hr at 4 °C and was applied to a Micro Bio-Spin column. The resin was washed with lysis buffer supplemented with 0.05 % DDM and 0.005 % CHS. EGFR was eluted from the resin with lysis buffer containing 0.16 mg/ml FLAG peptide, 0.05 % DDM and 0.005 % CHS. Protein concentration was estimated by the band intensities on a SDS-PAGE gel stained with Coomassie Brilliant Blue and by UV absorbance at 280 nm measured using NanoDrop (ThermoFisher).

### Western-blot assays using purified EGFR protein

~1 μM purified EGFR protein prepared as described above, was mixed with ~16 μM EGF or TGF-α in a buffer containing 25 mM Tris pH 8.0, 150 mM NaCl, 10 mM $MgCl_2$, 1 mM sodium orthovanadate, and 0.2 mM DTT. The mixture was incubated for 5 minutes, and phosphorylation reactions were initiated by addition of ATP to a final concentration of 1 mM. Reactions were typically carried out at 25 °C. At various time points, aliquots of the reaction solution were mixed with quenching solution, SDS-PAGE loading buffer supplemented with 20 mM EDTA, to achieve final concentration of 1 % SDS and 10 mM EDTA in the quenched solutions. The samples quenched at different time points were run on two 12 % acrylamide SDS-PAGE gels at 250 V, 400 mA for 35 min at room temperature.

The first SDS-PAGE gel was stained using Coomassie Brilliant Blue to monitor the total amount of EGFR protein loaded in each lane. The second SDS-PAGE gel was transferred to PVDF membrane using a semi-dry transfer apparatus TRANS-BLOT (Bio-Rad) at 25 V, 150 mA for 40 min at room temperature, in transfer buffer containing 25 mM Tris, 192 mM glycine, 0.02 % SDS, and 10 % Methanol. The PVDF membranes were blocked for 1 hr at room temperature in 5 % (w/v) dry milk dissolved in Tris-buffered saline with 0.1 % Tween-20 (TBST). A pan-phosphotyrosine antibody (pY20, catalog no. sc-12351, Santa Cruz Biotechnology) was diluted (1:2000) with 5 % (w/v) dry milk in TBST and incubated with the membranes at 4 °C overnight. The membranes were then washed three times with TBST, and incubated with horseradish peroxidase (HRP)-conjugated anti-mouse secondary antibody (catalog no. 7074, Cell Signaling Technology) at a 1:3000 dilution in TBST with 5 % dry milk at room temperature for 1 hr. The membranes were then washed three times with TBST, incubated with WesternBright Quantum chemiluminescent reagents (Advansta) for 2 min, and were imaged using ImageLab (Bio-Rad).

## Cell-based western-blot assays for EGFR autophosphorylation and Erk phosphorylation

### EGFR autophosphorylation in cells

N-terminal Flag-tagged full-length human EGFR was cloned into the pEGFP-N1 plasmid. HEK293T cells were transiently transfected with EGFR using polyethylenimine (PEI) in a six-well plate format. Twenty-four hrs after transfection, cells were serum starved overnight (~12 hr) in serum-free medium. For stimulation, cells were incubated with fresh medium without ligand, with EGF (16.7 nM), or with TGF-α (16.7 nM) for 1 hr at 4 °C (on ice). Following ligand stimulation, cells were pelleted and then immediately lysed in ice-cold cell lysis buffer (catalog no. 9803, Cell Signaling Technology) supplemented with cOmplete Protease Inhibitor Cocktail and PhosSTOP phosphatase inhibitors (Roche) for 30 min on ice. Cell lysates were clarified by centrifugation and western-blot analyses were carried out as described above for the purified EGFR proteins. A pan-phosphotyrosine antibody (pY20) was used to detect EGFR autophosphorylation. An anti-FLAG M2 antibody (catalog no. F1804, Sigma-Aldrich) was used to monitor the total amount of EGFR protein in the cell lysate loaded to each lane.

### Erk phosphorylation in cells

N-terminal Flag-tagged full-length human EGFR was cloned into the pEGFP-N1 plasmid. The mutations were introduced by QuikChange site-directed mutagenesis (Stratagene). Twenty-four hr after transfection, HEK293T cells transiently transfected with EGFR and EGFR variants were serum starved overnight (~12 hr) in serum-free medium. For stimulation, cells were incubated with fresh medium without ligand, with EGF (16.7 nM), or with TGF-α (16.7 nM) for 5 min at 37 °C. Following ligand stimulation, cells were immediately placed on ice and then lysed in ice-cold cell lysis buffer supplemented with cOmplete Protease Inhibitor Cocktail and PhosSTOP phosphatase inhibitors for 30 min on ice. Cell lysates were clarified by centrifugation and western blot analyses were carried out as described above. An anti-phospho-Erk antibody (catalog no. 4370, Cell Signaling Technology) is used to detect phosphorylation of p44 and p42 MAP kinase (Erk1 and Erk2). An anti-FLAG M2 antibody was used to monitor the total amount of EGFR protein in the cell lysate loaded to each lane.

All western blots were reproduced at least three times and the representative ones were presented.

## Acknowledgements

We thank the Cell Culture Facility at the University of California, Berkeley, for insect cell and mammalian cell cultures. We thank the Eric Gouaux group at the Vollum Institute at the Oregon Health & Science University, for the pEG_BacMam plasmid. This research was in part supported by the National Cryo-EM Facility of the National Cancer Institute at the Frederick National Laboratory for Cancer Research, under contract HSSN261200800001E. We also acknowledge the Berkeley Bay Area Cryo-EM Facility and the HHMI Janelia Cryo-EM Facility. Molecular graphics and analyses performed with UCSF Chimera, developed by the Resource for Biocomputing, Visualization, and Informatics at the University of California, San Francisco, with support from NIH P41-GM103311. This work used the Extreme Science and Engineering Discovery Environment (XSEDE), which is supported by National Science Foundation grant ACI-1548562. This work is in part supported by an award from the Canada Excellence Research Chair program (to SS) and by a philanthropic gift to the VGH and UBC Hospital Foundation (to SS).

## Additional information

### Competing interests

Deepti Karandur: DK is an early-career reviewer for eLife. Sriram Subramaniam: SS is Founder and Chief Executive Officer of Gandeeva Therapeutics Inc Reviewing editor, eLife. John Kuriyan: JK is a cofounder of Nurix Therapeutics and is on the scientific advisory boards of Carmot and Revolution Medicine. The other authors declare that no competing interests exist.

## Funding

| Funder | Grant reference number | Author |
|---|---|---|
| Canada Excellence Research Chairs, Government of Canada | | Sriram Subramaniam |
| VGH and UBC Hospital Foundation | | Sriram Subramaniam |

The funders had no role in study design, data collection and interpretation, or the decision to submit the work for publication.

## Author contributions

Yongjian Huang, Conceptualization, Data curation, Formal analysis, Investigation, Validation, Visualization, Writing – original draft, Writing – review and editing; Jana Ognjenovic, Data curation, Investigation, Writing – review and editing; Deepti Karandur, Data curation, Formal analysis, Writing – review and editing; Kate Miller, Data curation, Formal analysis; Alan Merk, Data curation; Sriram Subramaniam, Conceptualization, Funding acquisition, Supervision, Writing – review and editing; John Kuriyan, Conceptualization, Funding acquisition, Supervision, Writing – original draft, Writing – review and editing

## Author ORCIDs

Yongjian Huang ⓘ http://orcid.org/0000-0002-1168-6111
Deepti Karandur ⓘ http://orcid.org/0000-0002-6949-6337
Sriram Subramaniam ⓘ http://orcid.org/0000-0003-4231-4115
John Kuriyan ⓘ http://orcid.org/0000-0002-4414-5477

## Decision letter and Author response

Decision letter https://doi.org/10.7554/eLife.73218.sa1
Author response https://doi.org/10.7554/eLife.73218.sa2

# Additional files

## Supplementary files

• Transparent reporting form

## Data availability

Human EGFR protein sequence is available from UniProt accession no. P00533. The cryo-EM density maps of EGFR(WT):EGF complex in juxtaposed and separated conformations, EGFR(WT):TGF-α complex in juxtaposed and separated conformations, EGFR(L834R):EGF complex in juxtaposed and separated conformations, have been deposited to the Electron Microscopy Data Bank (EMDB) under the accession codes EMD-25522 and EMD-25523, EMD-25563 and EMD-25561, EMD-25558 and EMD-25559, respectively. The associated coordinates have been deposited to the PDB under accession codes 7SYD and 7SYE, 7SZ7 and 7SZ5, 7SZ0 and 7SZ1, respectively.

The following dataset was generated:

| Author(s) | Year | Dataset title | Dataset URL | Database and Identifier |
|---|---|---|---|---|
| Huang Y, Ognjenovic J, Karandur D, Miller K, Merk A, Subramaniam S, Kuriyan J | 2021 | Cryo-EM structure of the extracellular module of the full-length EGFR bound to EGF. "tips-juxtaposed" conformation | https://www.ebi.ac.uk/emdb/entry/EMD-25522 | EMDB, EMD-25522 |
| Huang Y, Ognjenovic J, Karandur D, Miller K, Merk A, Subramaniam S, Kuriyan J | 2021 | Cryo-EM structure of the extracellular module of the full-length EGFR bound to EGF. "tips-juxtaposed" conformation | https://www.rcsb.org/structure/7SYD | RCSB Protein Data Bank, 7SYD |

*Continued on next page*

*Continued*

| Author(s) | Year | Dataset title | Dataset URL | Database and Identifier |
|---|---|---|---|---|
| Huang Y, Ognjenovic J, Karandur D, Miller K, Merk A, Subramaniam S, Kuriyan J | 2021 | Cryo-EM structure of the extracellular module of the full-length EGFR bound to EGF. "tips-separated" conformation | https://www.ebi.ac.uk/emdb/entry/EMD-25523 | EMDB, EMD-25523 |
| Huang Y, Ognjenovic J, Karandur D, Miller K, Merk A, Subramaniam S, Kuriyan J | 2021 | Cryo-EM structure of the extracellular module of the full-length EGFR bound to EGF. "tips-separated" conformation | https://www.rcsb.org/structure/7SYE | RCSB Protein Data Bank, 7SYE |
| Huang Y, Ognjenovic J, Karandur D, Miller K, Merk A, Subramaniam S, Kuriyan J | 2021 | Cryo-EM structure of the extracellular module of the full-length EGFR bound to TGF-alpha. "tips-juxtaposed" conformation | https://www.ebi.ac.uk/emdb/entry/EMD-25563 | EMDB, EMD-25563 |
| Huang Y, Ognjenovic J, Karandur D, Miller K, Merk A, Subramaniam S, Kuriyan J | 2021 | Cryo-EM structure of the extracellular module of the full-length EGFR bound to TGF-alpha. "tips-juxtaposed" conformation | https://www.rcsb.org/structure/7SZ7 | RCSB Protein Data Bank, 7SZ7 |
| Huang Y, Ognjenovic J, Karandur D, Miller K, Merk A, Subramaniam S, Kuriyan J | 2021 | Cryo-EM structure of the extracellular module of the full-length EGFR bound to TGF-alpha. "tips-separated" conformation | https://www.ebi.ac.uk/emdb/entry/EMD-25561 | EMDB, EMD-25561 |
| Huang Y, Ognjenovic J, Karandur D, Miller K, Merk A, Subramaniam S, Kuriyan J | 2021 | Cryo-EM structure of the extracellular module of the full-length EGFR bound to TGF-alpha. "tips-separated" conformation | https://www.rcsb.org/structure/7SZ5 | RCSB Protein Data Bank, 7SZ5 |
| Huang Y, Ognjenovic J, Karandur D, Miller K, Merk A, Subramaniam S, Kuriyan J | 2021 | Cryo-EM structure of the extracellular module of the full-length EGFR L834R bound to EGF. "tips-juxtaposed" conformation | https://www.ebi.ac.uk/emdb/entry/EMD-25558 | EMDB, EMD-25558 |
| Huang Y, Ognjenovic J, Karandur D, Miller K, Merk A, Subramaniam S, Kuriyan J | 2021 | Cryo-EM structure of the extracellular module of the full-length EGFR L834R bound to EGF. "tips-juxtaposed" conformation | https://www.rcsb.org/structure/7SZ0 | RCSB Protein Data Bank, 7SZ0 |
| Huang Y, Ognjenovic J, Karandur D, Miller K, Merk A, Subramaniam S, Kuriyan J | 2021 | Cryo-EM structure of the extracellular module of the full-length EGFR L834R bound to EGF. "tips-separated" conformation | https://www.ebi.ac.uk/emdb/entry/EMD-25559 | EMDB, EMD-25559 |
| Huang Y, Ognjenovic J, Karandur D, Miller K, Merk A, Subramaniam S, Kuriyan J | 2021 | Cryo-EM structure of the extracellular module of the full-length EGFR L834R bound to EGF. "tips-separated" conformation | https://www.rcsb.org/structure/7SZ1 | RCSB Protein Data Bank, 7SZ1 |

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
