## [Editor Report]

This is an impressive study providing solid evidence for a molecular mechanism by which two related, high-affinity growth factors, binding in exactly the same site, can achieve differential signaling outputs through a dimerized receptor tyrosine kinase, and represents an important advance in the field.

---

## [Decision Letter]

**Decision letter after peer review:**

Thank you for submitting your article "A molecular mechanism for the generation of ligand-dependent differential outputs by the epidermal growth factor receptor" for consideration by *eLife*. Your article has been reviewed by 3 peer reviewers, and the evaluation has been overseen by a Reviewing Editor and Jonathan Cooper as the Senior Editor. The following individuals involved in review of your submission have agreed to reveal their identity: Stevan R Hubbard (Reviewer #1); Xiao-chen Bai (Reviewer #3).

Essential revisions:

1. Cryo-EM 3D classification was widely used to analyze the dynamics of protein complexes. It could be efficient in probing new intermediate states. But it is not strictly a quantitative method unless certain "unbiased" procedures are applied throughout the complete workflow of image processing. For example, the authors only showed classification results of one dataset (Figure 2). The particles appeared to be evenly distributed in the ten classes, which was a sign that the classification was not efficient and the resulting reconstructions are still mixtures of multiple states. To support the main conclusion of the manuscript, the authors should show the classification results of all datasets: (1) To our knowledge, cryoSparc is less efficient in separating particles into different conformations compared to Relion. The authors could try Relion and multiple rounds of 3D classification could be implemented. (2) Mask-based classification should be applied. (3) The authors could provide an analysis on the changes of particle distribution among different datasets.

2. Although the mutations introduced into EGFR (W492G, etc.) that mitigate the phosphorylation differences between EGF and TGF-αt are strong evidence for the authors' hypothesis, it is important to show (through a ligand-concentration series) that saturating amounts of TGF-α cannot achieve the same EGFR phosphorylation level as saturating amounts of EGF. The authors used a single concentration of EGF or TGF-α (16 µM in vitro, 16 nM in-cell) in the results shown in Figure 9 and Figure 9-supplement 1. Any differences in affinity and/or ligand solubility/delivery, etc. between EGF and TGF-α could explain the difference in phosphorylation levels in the experiments with wild-type EGFR.

3. For the cell-based phosphorylation results shown in Figure 9C and E, the authors have already performed at least three biological replicates. It would be useful for the readers if the authors could include densitometry analysis and appropriate statistics on the biological replicates to show the effect of mutations and different types of ligand binding on EGFR activity in a more quantitative way.

4. What is rationale to use 16.7 nM as the ligand concentration in the cell-based assay? Has this been commonly used in EGFR activity assay in previous work? Have the authors tried other ligand concentration?

5. Related to previous two points, have the authors tested the effect of L834R mutation on the activity of EGFR using similar cell-based phosphorylation experiment? This is just to confirm that L834R is indeed a gain-of-function mutation. It also would be interesting to check whether EGFR-L834R also has different activities when responding to EGF or TGF-α.

6. The authors need to prepare similar figures as Figure3A to show the cryo-EM image processing workflows for EGFR/TGF-α and EGFR-L834R/EGF datasets.

7. The authors claimed that conformations between ECD, TM and ICD are coupled within ligand-bound EGFR. But the TM and kinase domains are not resolved at all in any of the classes. Could the authors provide a brief explanation for this issue?

*Reviewer #1:*

The manuscript by Huang et al. reports the cryo-EM structures of EGF and TGFalpha bound to full-length EGFR. As for other receptor tyrosine kinases like the insulin and IGF1 receptors, the transmembrane helices and cytoplasmic kinase domains are not resolved in the cryo-EM maps.

3D classification of EGF-EGFR revealed multiple, closely-related conformational states of the ligand-bound ectodomain, in which a "scissor-like" rotation of the EGF binding portion of the ectodomain (D1-3) was correlated with a separation of the ends of the membrane-proximal domain (D4); the larger the scissor angle (~25{degree sign}), the closer the ends of D4 (~5 Å), and vice versa. For the smaller scissor angle of ~10{degree sign}, the two-fold symmetry of the EGF+EGFR complex breaks down, such that one of the D4 domains pivots from D1-3 further than the other one, resulting in a D4 separation of ~20 Å.

The authors utilized previous NMR data on the isolated TM helices of EGFR, which indicated that there are two mutually exclusive crossovers points between the TM helices, one closer to the N-termini of the helices and one closer to the C-termini. Molecular dynamics simulations performed by the authors showed that, in general, the "tips-separated" configuration of the D4 domains was correlated with the N-terminal apposition of the TM helices, and the "tips-juxtaposed" configuration was correlated with the C-terminal apposition.

Previous biochemical data had indicated/suggested that the N-terminal dimerization of the TM helices results in higher kinase activity (through formation of the asymmetric kinase dimer) than the C-terminal dimerization, even though the C-terminal dimerization places the cytoplasmic juxtamembrane (JM) regions (leading into the kinase domains) closer together.

The authors determined cryo-EM structures of EGF bound to an EGFR mutant, L834R, which is a gain-of-function substitution in the activation loop of the kinase domain, and found that D4 in the tips-separated conformation was stabilized vs. in wild-type EGFR, indicating that a stabilized asymmetric kinase dimer is conformationally coupled to the tips-separated ectodomain conformation.

The authors determined cryo-EM structures with TGFalpha bound to EGFR and found that, in this ensemble of structures, D4 in the tips-separated conformation was destabilized (vs. in the EGF-EGFR structures) because of slight differences in the ligand-binding head of EGFR induced by TGFalpha vs. EGF binding.

All of these data - theirs and others - led to the hypothesis that EGF is a higher activity ligand than TGFalpha (despite both being high-affinity binders to EGFR) because of the conformational coupling between the ligand-binding head of EGFR, the distal tips of D4, the TM helices, the cytoplasmic JM region, and the asymmetric kinase dimer. To test this hypothesis, the authors performed in vitro and in-cell activity assays and, indeed, found that the level of EGFR phosphorylation was higher when stimulated with EGF vs. TGFalpha.

To provide evidence that the conformational coupling described above was responsible, the authors generated mutant EGFRs -a point mutation in D4 (W492G) and insertion in and replacement of the extracellular JM region - and measured phosphorylation levels upon stimulation with EGF or TGFalpha. These data showed that increasing the flexibility in these regions (through mutation) abrogated the phosphorylation difference in the two cases (EGF vs. TGFalpha), consistent with their hypothesis.

In summary, this is an impressive study providing solid evidence for a molecular mechanism by which two related, high-affinity growth factors, binding in exactly the same site, can achieve differential signaling outputs through a dimerized receptor tyrosine kinase, and represents an important advance in the field.

Although it is surprising that the small conformational differences in the ligand-binding head of EGFR resulting from either EGF or TGFalpha binding can be "faithfully" propagated through the D4 domains into the TM helices and then into the cytoplasmic region to affect asymmetric kinase dimer formation, the data are quite convincing, especially the mutagenesis data.

Specific Points:

1) Although the mutations introduced into EGFR (W492G, etc.) that mitigate the phosphorylation differences between EGF and TGFalpha are strong evidence for the authors' hypothesis, it is important to show (through a ligand-concentration series) that saturating amounts of TGFalpha cannot achieve the same EGFR phosphorylation level as saturating amounts of EGF. The authors used a single concentration of EGF or TGFalpha (16 µM in vitro, 16 nM in-cell) in the results shown in Fig. 9 and Fig. 9-supplement 1. Any differences in affinity and/or ligand solubility/delivery, etc. between EGF and TGFalpha could explain the difference in phosphorylation levels in the experiments with wild-type EGFR.

2) In Fig. 2, it would be instructive for the reader to know exactly what the high and low map contour levels are (e.g., number of sigmas).

Other concerns:

1. As stated in line 266, the EM density for one domain IV leg in the tips-separated conformation of EGFR: TGF-α was poorly defined. The residual EM density seemed to suggested a "juxtaposed" conformation rather than "separated" state in in EGFR(L834R):EGF and EGFR(L834R):EGF (Figure 7, top panels).

2. Line 197-201. The MD simulation could be done with the TM region plus extracellular module.

3. Figure 9, panel E does not have a WT control.

*Reviewer #2:*

EGFR can be activated by several extracellular ligands. The molecular mechanisms of EGFR in differentiating extracellular signals from these ligands and transforming them into distinct intracellular signaling outputs are not fully understood. In this manuscript, Huang et al. carried out structural analysis of the full-length human EGFR (with ligand EGF or TGF-α) using cryo-EM and MD simulation. The authors reported that the dimeric structure of the two extracellular modules is not rigid at the dimeric interface, resulting in conformational fluctuations of individual domains. One interesting observation was the membrane-proximal tip of the extracellular module in representative two conformations, "separated" and "juxtaposed" states. The authors next tried to correlate the structural dynamics of EGFR to its signaling outputs.

Cryo-EM 3D classification was widely used to analyze the dynamics of protein complexes. It could be efficient in probing new intermediate states. But it is not strictly a quantitative method unless certain "unbiased" procedures are applied throughout the complete workflow of image processing. For example, the authors only showed classification results of one dataset (Figure 2). The particles appeared to be evenly distributed in the ten classes, which was a sign that the classification was not efficient and the resulting reconstructions are still mixtures of multiple states. To support the main conclusion of the manuscript, the authors should show the classification results of all datasets: (1) To our knowledge, cryoSparc is less efficient in separating particles into different conformations compared to Relion. The authors could try Relion and multiple rounds of 3D classification could be implemented. (2) Mask-based classification should be applied. (3) The authors could provide an analysis on the changes of particle distribution among different datasets.

*Reviewer #3:*

This will be a landmark work in the RTK and EGFR fields. Huang et. al reported a series of cryo-EM structures of full-length EGFR/EGF complexes in different conformations. The major difference among these structures is the distance between the membrane proximal domains IV of EGFR. Although the TM and kinase domains of EGFR were not resolved in the cryo-EM maps, through comprehensive structural analysis and MD simulations, the authors proposed that, the EGFR/EGF complex with separated domains IV would induce N-terminal associated dimeric TM domain and high activity; whereas the EGFR/EGF complex with juxtaposed domains IV would promote C-terminal associated dimeric TM domain and low activity. Such claim is strongly supported by two structure evidences: (1) In the cryo-EM structure of EGFR L834R mutant/EGF complex (a mutant that is supposed to have higher activity than EGFR WT), the separated domains IV is captured in a more stable state. (2) In the cryo-EM structure of EGFR with a weaker ligand TGF-a bound, the separated domains IV is in a more flexible conformation. In addition, the authors also introduced some mutations to EGFR, designed to break the structural coupling between domain IV and TM domain. These EGFR mutants can't response to EGF and TGF-a differently, which further supports the major conclusion of this work that the conformation of ECD determines the conformation of TM as well as the downstream signaling. Overall, the experiments were well designed, and the structural and functional works are of great quality.

---

## [Author Response]

Essential revisions:1. Cryo-EM 3D classification was widely used to analyze the dynamics of protein complexes. It could be efficient in probing new intermediate states. But it is not strictly a quantitative method unless certain "unbiased" procedures are applied throughout the complete workflow of image processing. For example, the authors only showed classification results of one dataset (Figure 2). The particles appeared to be evenly distributed in the ten classes, which was a sign that the classification was not efficient and the resulting reconstructions are still mixtures of multiple states. To support the main conclusion of the manuscript, the authors should show the classification results of all datasets: (1) To our knowledge, cryoSparc is less efficient in separating particles into different conformations compared to Relion. The authors could try Relion and multiple rounds of 3D classification could be implemented. (2) Mask-based classification should be applied. (3) The authors could provide an analysis on the changes of particle distribution among different datasets.

We agree with the reviewer on that the current methods for cryo-EM 3D classification are not necessarily very definitive in terms of determining the dynamics of protein complexes. In this regard, it is important to note that the classification procedure resulted in the identification of alternative conformations of the ligand-binding head that correspond closely to structures that had been identified previously in crystal structures of different states of the receptor. Specifically, as noted on page 6 of the manuscript (starting line 123), the “tips-juxtaposed” conformation is almost precisely the same as that seen in a crystal structure of the isolated EGFR extracellular module bound to EGF (PDB code: 3NJP). Likewise, the “tips-separated” conformation corresponds to the structure of a truncated form of the extracellular module of EGFR (ligand-binding head only) bound to TGF-α (PDB code: 1MOX). While we agree with the point made by the reviewer, the important point is that this correspondence to previously determined crystal structures makes the key conclusions of our cryo-EM analysis robust, although a quantitative determination of conformational populations cannot be obtained reliably.

We have now added the classification results for the other two datasets as well, L834R:EGF (Figure 3—figure supplement 3A) and EGFR:TGF-α (Figure 3—figure supplement 3B).

In response to reviewer’s comments (1) and (2), despite intense efforts, including multiple rounds of mask-based 3D classifications and partial signal subtraction, we were unable to achieve better classification results for our datasets using Relion.

Regarding comment (3), the difference in the particle distributions among different conformational states are quite small, when the EGFR:EGF, EGFR:TGF-α, and L834R:EGF complexes are compared. Specifically, the ratio between the particle numbers of the juxtaposed and of the separated conformations is ~1.18 (105,728: 89,757), ~1.24 (126,471: 102,206), and ~1.23 (123,670: 100,615), in the EGFR:EGF, EGFR:TGF-α, and L834R:EGF complexes, respectively. We interpret this to mean that the principal effect of changing the ligand, or introducing an activating mutation, is in the altered dynamics of the Domain IV leg in the tips-separated conformation.

2. Although the mutations introduced into EGFR (W492G, etc.) that mitigate the phosphorylation differences between EGF and TGF-α are strong evidence for the authors' hypothesis, it is important to show (through a ligand-concentration series) that saturating amounts of TGF-α cannot achieve the same EGFR phosphorylation level as saturating amounts of EGF. The authors used a single concentration of EGF or TGF-α (16 µM in vitro, 16 nM in-cell) in the results shown in Figure 9 and Figure 9-supplement 1. Any differences in affinity and/or ligand solubility/delivery, etc. between EGF and TGF-α could explain the difference in phosphorylation levels in the experiments with wild-type EGFR.

This is a very important point raised by reviewer, and it is critical to demonstrate that the phosphorylation differences between EGF and TGF-α complexes are achieved by saturating amounts of both EGF and TGF-α.

The affinities of EGFR for EGF and TGF-α were reported to be ~0.6 nM and ~3 nM (Macdonald-Obermann and Pike, 2014), respectively. In cell-based assays, we used saturating concentrations of EGF and TGF-α (100 ng/ml or 16.7 nM), which have been commonly used in EGFR activity assays in previous work (Macdonald-Obermann and Pike, 2014; Sinclair et al., 2018). For in vitro assay using purified full-length EGFR protein, EGF and TGF-α were added in 16 times (16 μM) excess of EGFR (1 μM).

To further demonstrate that TGF-α cannot achieve the same EGFR activation level as EGF, even at concentrations that are higher than the saturating concentration (100 ng/ml), we conducted a new set of cell-based experiments to measure the phosphorylation levels of Erk in cells that are transiently transfected with wild-type EGFR. As shown in Figure 9—figure supplement 1C, EGF stimulates higher levels of Erk phosphorylation than TGF-α does, regardless of the ligand concentrations used.

3. For the cell-based phosphorylation results shown in Figure 9C and E, the authors have already performed at least three biological replicates. It would be useful for the readers if the authors could include densitometry analysis and appropriate statistics on the biological replicates to show the effect of mutations and different types of ligand binding on EGFR activity in a more quantitative way.

We thank the reviewer for this suggestion. We have now included the suggested densitometry analysis of the cell-based EGFR activity assays as the Figure 9—figure supplement 1E.

4. What is rationale to use 16.7 nM as the ligand concentration in the cell-based assay? Has this been commonly used in EGFR activity assay in previous work? Have the authors tried other ligand concentration?

Please see point #2, above.

5. Related to previous two points, have the authors tested the effect of L834R mutation on the activity of EGFR using similar cell-based phosphorylation experiment? This is just to confirm that L834R is indeed a gain-of-function mutation. It also would be interesting to check whether EGFR-L834R also has different activities when responding to EGF or TGF-α.

We have conducted the cell-based experiment to test the effect of L834R mutation on the activity of EGFR.

As shown in Figure 9—figure supplement 1D, in comparison to the wild-type EGFR, L834R mutation appears to mildly increase the phosphorylation level of Erk in the presence of either ligands, which is consistent with its role as a gain-of-function mutation suggested by previous studies (Red Brewer et al., 2013; Shan et al., 2012).

6. The authors need to prepare similar figures as Figure3A to show the cryo-EM image processing workflows for EGFR/TGF-α and EGFR-L834R/EGF datasets.

We have now added the figures for the cryo-EM image processing workflows of EGFR(L834R):EGF and EGFR:TGF-α datasets, as Figure 3—figure supplement 3A and Figure 3—figure supplement 3B.

7. The authors claimed that conformations between ECD, TM and ICD are coupled within ligand-bound EGFR. But the TM and kinase domains are not resolved at all in any of the classes. Could the authors provide a brief explanation for this issue?

Based on our current structural model, the coupling between the extracellular module and the transmembrane helices does not require a rigid connection between the two domains. Instead, it is the extent of the proximity of the tips of the extracellular domains in a dimer, at the point where they connect to the transmembrane helices, that is the central structural feature responsible for coupling between the conformations of the extracellular module and the transmembrane helices. We were unable to resolve the transmembrane and the intracellular module of the receptor, even within the subclasses of the separated or juxtaposed conformations, presumably due to major segmental flexibility observed between the extracellular module and the transmembrane helices. This flexible connection between the extracellular module and the transmembrane helices is likely caused by the flexible linker connecting the extracellular module to the transmembrane helices. And we do not expect this flexible linker to become rigid in any of the subclasses. But we do expect the conformational heterogeneity within the transmembrane helices and the intracellular module to get reduced when they are linked to the tips-separated conformation of the extracellular module.

It is worth noting that recent studies on other full-length receptor tyrosine kinases all failed to resolve the transmembrane domains or intracellular modules, likely due to similar flexibility (Li et al., 2019; Uchikawa et al., 2019; Zhang et al., 2020; Diwanji et al., 2021).

Reviewer #1:The manuscript by Huang et al. reports the cryo-EM structures of EGF and TGFalpha bound to full-length EGFR. As for other receptor tyrosine kinases like the insulin and IGF1 receptors, the transmembrane helices and cytoplasmic kinase domains are not resolved in the cryo-EM maps.3D classification of EGF-EGFR revealed multiple, closely-related conformational states of the ligand-bound ectodomain, in which a "scissor-like" rotation of the EGF binding portion of the ectodomain (D1-3) was correlated with a separation of the ends of the membrane-proximal domain (D4); the larger the scissor angle (~25{degree sign}), the closer the ends of D4 (~5 Å), and vice versa. For the smaller scissor angle of ~10{degree sign}, the two-fold symmetry of the EGF+EGFR complex breaks down, such that one of the D4 domains pivots from D1-3 further than the other one, resulting in a D4 separation of ~20 Å.The authors utilized previous NMR data on the isolated TM helices of EGFR, which indicated that there are two mutually exclusive crossovers points between the TM helices, one closer to the N-termini of the helices and one closer to the C-termini. Molecular dynamics simulations performed by the authors showed that, in general, the "tips-separated" configuration of the D4 domains was correlated with the N-terminal apposition of the TM helices, and the "tips-juxtaposed" configuration was correlated with the C-terminal apposition.Previous biochemical data had indicated/suggested that the N-terminal dimerization of the TM helices results in higher kinase activity (through formation of the asymmetric kinase dimer) than the C-terminal dimerization, even though the C-terminal dimerization places the cytoplasmic juxtamembrane (JM) regions (leading into the kinase domains) closer together.The authors determined cryo-EM structures of EGF bound to an EGFR mutant, L834R, which is a gain-of-function substitution in the activation loop of the kinase domain, and found that D4 in the tips-separated conformation was stabilized vs. in wild-type EGFR, indicating that a stabilized asymmetric kinase dimer is conformationally coupled to the tips-separated ectodomain conformation.The authors determined cryo-EM structures with TGFalpha bound to EGFR and found that, in this ensemble of structures, D4 in the tips-separated conformation was destabilized (vs. in the EGF-EGFR structures) because of slight differences in the ligand-binding head of EGFR induced by TGFalpha vs. EGF binding.All of these data - theirs and others - led to the hypothesis that EGF is a higher activity ligand than TGFalpha (despite both being high-affinity binders to EGFR) because of the conformational coupling between the ligand-binding head of EGFR, the distal tips of D4, the TM helices, the cytoplasmic JM region, and the asymmetric kinase dimer. To test this hypothesis, the authors performed in vitro and in-cell activity assays and, indeed, found that the level of EGFR phosphorylation was higher when stimulated with EGF vs. TGFalpha.To provide evidence that the conformational coupling described above was responsible, the authors generated mutant EGFRs -a point mutation in D4 (W492G) and insertion in and replacement of the extracellular JM region - and measured phosphorylation levels upon stimulation with EGF or TGFalpha. These data showed that increasing the flexibility in these regions (through mutation) abrogated the phosphorylation difference in the two cases (EGF vs. TGFalpha), consistent with their hypothesis.In summary, this is an impressive study providing solid evidence for a molecular mechanism by which two related, high-affinity growth factors, binding in exactly the same site, can achieve differential signaling outputs through a dimerized receptor tyrosine kinase, and represents an important advance in the field.Although it is surprising that the small conformational differences in the ligand-binding head of EGFR resulting from either EGF or TGFalpha binding can be "faithfully" propagated through the D4 domains into the TM helices and then into the cytoplasmic region to affect asymmetric kinase dimer formation, the data are quite convincing, especially the mutagenesis data.Specific Points:1) Although the mutations introduced into EGFR (W492G, etc.) that mitigate the phosphorylation differences between EGF and TGFalpha are strong evidence for the authors' hypothesis, it is important to show (through a ligand-concentration series) that saturating amounts of TGFalpha cannot achieve the same EGFR phosphorylation level as saturating amounts of EGF. The authors used a single concentration of EGF or TGFalpha (16 µM in vitro, 16 nM in-cell) in the results shown in Fig. 9 and Fig. 9-supplement 1. Any differences in affinity and/or ligand solubility/delivery, etc. between EGF and TGFalpha could explain the difference in phosphorylation levels in the experiments with wild-type EGFR.

Please see Essential Revisions point #2, above.

2) In Fig. 2, it would be instructive for the reader to know exactly what the high and low map contour levels are (e.g., number of sigmas).

We have now included the values of sigmas (map contour levels) in the Figure 2 legend.

Other concerns:1. As stated in line 266, the EM density for one domain IV leg in the tips-separated conformation of EGFR: TGF-α was poorly defined. The residual EM density seemed to suggested a "juxtaposed" conformation rather than "separated" state in in EGFR(L834R):EGF and EGFR(L834R):EGF (Figure 7, top panels).

Based on the small intersubunit rotational angle observed in the ligand-binding head region, this density (Figure 7, top right panel) corresponds to the tip-separated conformation of EGFR:TGF-a complex. The residual EM density for the domain IV leg (Figure 7, top right panel) is very sparse and noisy. At a lower contour, the same map clearly shows that the domain IV leg is disordered, rather than adopting the tips-juxtaposed position.

2. Line 197-201. The MD simulation could be done with the TM region plus extracellular module.

This is a good suggestion for future work. MD simulations of a system that includes the extracellular module, the transmembrane helices, and the membrane, requires an investment of time beyond the scope of the current work.

3. Figure 9, panel E does not have a WT control.

Due to the low throughput and time-sensitive nature of these cell-based assays, wild-type samples were not always included with every mutants sample on the same western blots. At least three independent biological replicates have been done for each mutants as well as the wild-type EGFR. We have now included a densitometry analysis of these cell-based EGFR activity assays to show the effects of mutations on EGFR activity stimulated by EGF and TGF-a (Figure 9-figure supplement 1E).

1) Although the mutations introduced into EGFR (W492G, etc.) that mitigate the phosphorylation differences between EGF and TGF-α are strong evidence for the authors' hypothesis, it is important to show (through a ligand-concentration series) that saturating amounts of TGF-α cannot achieve the same EGFR phosphorylation level as saturating amounts of EGF. The authors used a single concentration of EGF or TGF-α (16 µM in vitro, 16 nM in-cell) in the results shown in Figure 9 and Figure 9-supplement 1. Any differences in affinity and/or ligand solubility/delivery, etc. between EGF and TGF-α could explain the difference in phosphorylation levels in the experiments with wild-type EGFR.

Please see Essential Revisions point #2, above.

2) In Figure 2, it would be instructive for the reader to know exactly what the high and low map contour levels are (e.g., number of sigmas).

We have now included the values of sigmas (map contour levels) in the Figure 2 legend.

Reviewer #2:EGFR can be activated by several extracellular ligands. The molecular mechanisms of EGFR in differentiating extracellular signals from these ligands and transforming them into distinct intracellular signaling outputs are not fully understood. In this manuscript, Huang et al. carried out structural analysis of the full-length human EGFR (with ligand EGF or TGF-α) using cryo-EM and MD simulation. The authors reported that the dimeric structure of the two extracellular modules is not rigid at the dimeric interface, resulting in conformational fluctuations of individual domains. One interesting observation was the membrane-proximal tip of the extracellular module in representative two conformations, "separated" and "juxtaposed" states. The authors next tried to correlate the structural dynamics of EGFR to its signaling outputs.Cryo-EM 3D classification was widely used to analyze the dynamics of protein complexes. It could be efficient in probing new intermediate states. But it is not strictly a quantitative method unless certain "unbiased" procedures are applied throughout the complete workflow of image processing. For example, the authors only showed classification results of one dataset (Figure 2). The particles appeared to be evenly distributed in the ten classes, which was a sign that the classification was not efficient and the resulting reconstructions are still mixtures of multiple states. To support the main conclusion of the manuscript, the authors should show the classification results of all datasets: (1) To our knowledge, cryoSparc is less efficient in separating particles into different conformations compared to Relion. The authors could try Relion and multiple rounds of 3D classification could be implemented. (2) Mask-based classification should be applied. (3) The authors could provide an analysis on the changes of particle distribution among different datasets.

Please see Essential Revisions point #1, above.